# Incorporating experimentally derived streamflow contributions into model parameterization to improve discharge prediction

Andreas Hartmann[1, 2, 3], Jean-Lionel Payeur-Poirier[4], Luisa Hopp[4]

[1] Institute of Groundwater Management, Technical University of Dresden, 01069 Dresden, Germany
[2] Chair of Hydrological Modeling and Water Resources, University of Freiburg, Germany
[3] Department of Civil Engineering, University of Bristol, United Kingdom
[4] Department of Hydrology, Bayreuth Center of Ecology and Environmental Research (BayCEER), University of Bayreuth, Bayreuth, Germany

*Correspondence to*: Andreas Hartmann (andreas.hartmann@tu-dresden.de)

**Abstract.** Environmental tracers have been used to separate streamflow components for many years. They allow to quantify the contribution of water originating from different sources such as direct runoff from precipitation, subsurface stormflow or groundwater to total streamflow at variable flow conditions. Although previous studies have explored the value of incorporating experimentally derived fractions of event and pre-event water into hydrological models, a thorough analysis of the value of incorporating hydrograph separation derived information on multiple streamflow components at varying flow conditions into model parameter estimation has not yet been performed. This study explores the value of such information to achieve more realistic simulations of catchment discharge. We use a modified version of the process-oriented HBV model that simulates catchment discharge through the interplay of hillslope, riparian zone discharge and groundwater discharge at a small forested catchment which is located in the mountainous north of South Korea subject to a monsoon season between June and August. Applying a Monte Carlo based parameter estimation scheme and the Kling Gupta efficiency (KGE) to compare discharge observations and simulations across two seasons (2013 & 2014), we show that the model is able to provide accurate simulations of catchment discharge (KGE $\geq$ 0.8) but fails to provide robust predictions and realistic estimates of the contribution of the different streamflow components. Using a simple framework that compares simulated and observed contributions of hillslope, riparian zone and groundwater to total discharge during two sub-periods, we show that the precision of simulated streamflow components can be increased while remaining with accurate discharge simulations. We further show that the additional information increases the identifiability of all model parameters and results in more robust predictions. Our study shows how tracer derived information on streamflow contributions can be used to improve the simulation and predictions of streamflow at the catchment scale without adding additional complexity to the model. The complementary use of temporally resolved observations of streamflow components and modelling provides a promising direction to improve discharge prediction by representing model internal dynamics more realistically.

## 1 Introduction

At many catchments, particularly in temperate regions, subsurface stormflow (SSF) is an important event-scale mechanism of streamflow generation (Chifflard et al., 2019; Bachmair and Weiler, 2011; Blume et al., 2016; Barthold and Woods, 2015). SSF often occurs at hillslopes with contrasting soil hydraulic properties within the soil profile favouring lateral flow rather than vertical percolation of infiltrating waters or where rising groundwater levels reach more permeable layers of the soil (Bishop et al., 1990). Previous work has shown that SSF can be an important component of runoff generation at the catchment scale (Zillgens et al., 2007), adding to flood generation (Markart et al., 2015), or nutrient and contaminant transport (Zhao et al., 2013). The experimental investigation of SSF requires intensive instrumentation, and therefore only few studies have attempted to directly measure SSF on natural hillslopes (Freer et al., 2002; Tromp-Van Meerveld and McDonnell, 2006; Du et al., 2016; Woods and Rowe, 1996). If direct field observations of SSF are not possible, sampling and characterizing subsurface water using tracers (soil water, shallow groundwater) can be a way forward to evaluate the relevance of SSF for streamflow generation. The tracer signatures of different water source areas or flow pathways (also called end-members) are used to compute in a mass balance approach the potential relative contributions of the sampled water sources required to result in the observed tracer signals in streamflow. Other than early approaches that split streamflow into event and pre-event water (Sklash et al., 1979; Kendall et al., 2001),  these approaches rely on the assumption that streamflow is a mixture of distinct water sources within the catchment. This hydrograph separation technique and more advanced multivariate statistical tools for comprehensive data sets, such as the End Member Mixing Analysis employing a principal component analysis, have extensively been used in streamflow generation studies (Brown et al., 1999; Christophersen and Hooper, 1992; Burns et al., 2001; Inamdar et al., 2013). However, the initiation, pathways, residence times, quantity, or spatial origin of SSF in various landscapes are still poorly understood. Due to this lack of a general understanding of the occurrence of and controls on SSF, only few modelling studies focus on the realistic simulation of SSF (Chifflard et al., 2019; Hopp and McDonnell, 2009; Appels et al., 2015).

Conceptual models lump together the spatial heterogeneity of hydrological properties of entire catchments or hydrotopes while still considering dominant hydrological processes (Wagener and Gupta, 2005). Different streamflow components and catchment internal fluxes are usually represented by the outflows of simple or modified linear reservoirs: For instance, the HBV model (Hydrologiska Byrans Vattenavdelning) (Lindström et al., 1997; Seibert and Vis, 2012) represents the interplay between subsurface stormflow and groundwater by a shallow groundwater reservoir with two outlets. When below a predefined threshold, only one outlet provides discharge to the stream. But when exceeding the threshold, the more dynamic second outlet releases additional water, which is one way of representing the "fill and spill" dynamics of SSF observed by (Tromp-Van Meerveld and McDonnell, 2006). A similar procedure is used in the TOPMODEL (Beven and Kirkby, 1979; Clark et al., 2008) or the Precipitation Runoff Modeling System (PRMS, (Leavesley et al., 1983; Markstrom et al., 2015) that uses a threshold to initiate subsurface stormflow (referred to by "preferential flow" in the model's manual). Physically based models usually discretize the catchment into a grid of rectangular or triangular cells and apply physical equations, e.g., Richards equation or

the groundwater flow equations, on each of them individually. That way they provide spatially distributed information on the flow and storage behaviour of the simulated catchments. Similar to conceptual models, many physically based models consider contributions of different water sources, e.g., direct input of precipitation, subsurface stormflow or groundwater, to total catchment discharge. For instance, the WaSiM-ETH model (Schulla and Jasper, 2007) considers subsurface stormflow by calculating interflow from hydraulic conductivity, river density, soil moisture and the matric potential. The SWAT model (Neitsch et al., 2011) uses a kinematic storage model to consider interflow, or the LARSIM model (Bremicker, 2000) uses the saturation deficit of the soil and a lateral drainage parameter to calculate subsurface stormflow.

In order to represent SSF correctly within conceptual and physically based models, the model parameters controlling the initiation and rate of SSF have to be estimated. However, in most of the model applications, little information about SSF model parameters is available and modellers have to rely on inverse parameter assessment approaches (Vrugt et al., 2008). Due to the limited information content of discharge (Wheater et al., 1986; Ye et al., 1997), the distinction of model internal lateral flow paths like surface runoff, SSF, groundwater, etc., remains uncertain (Seibert and McDonnell, 2002). Previous work already used field observations in addition to discharge to confine model parameters and simulated processes using, for instance, hydrochemical information (Kuczera and Mroczkowski, 1998; Hartmann et al., 2017; Uhlenbrook and Leibundgut, 1999) and stable water isotopes (Yang et al., 2021; Mayer-Anhalt et al., 2022; Sprenger et al., 2015). The use of stable water isotopes in conceptual models resulted in a better quantification of the passive catchments storage (Birkel et al., 2011) and increased parameter identifiably at humid test sites in Scotland (Birkel et al., 2014), while other studies showed the usefulness of isotopes and hydrochemical information for model structure identification (Capell et al., 2012; McMillan et al., 2012; Hartmann et al., 2013). Generally, the inclusion of environmental tracers resulted in better (multi-variate) model calibration and validation, especially at larger scales(Holmes et al., 2022; Stadnyk et al., 2013; Bergström et al., 2002), which is further elaborated on in a review on approaches for tracer-aided modelling is provided by Birkel and Soulsby (2015). In a multi-objective approach, (Seibert and McDonnell, 2002) showed that the inclusion of groundwater observations and discontinuous observations of event water contributions derived from hydrograph separation allowed for an improved confinement of simulated processes. However, a detailed analysis of the usefulness of incorporating more detailed information of experimentally derived streamflow components is, to our knowledge, not yet available.

This study explores the value of experimentally derived contributions to streamflow to identify the increase in accuracy of simulated streamflow components at the catchment scale. We use a modified version of the process-oriented HBV model and Monte Carlo based parameter estimation framework to (1) obtain acceptable simulations of total streamflow at the catchment outlet and (2) incorporate experimentally derived information on the contributions of water originating from the hillslope, the riparian zone and from groundwater to total streamflow into model parameter estimation. By iteratively adding this information to the parameter estimation, we can quantify the impact of the additional data on parameter identifiability and on the uncertainty of discharge simulations during variable flow conditions. We apply our approach at a well-instrumented test site in the monsoonal mountainous north of South Korea during two consecutive seasons.

## 2 Experimental work and hydrograph separation

### 2.1 Test catchment

Our test catchment is located in a mountainous area in the northeast of South Korea (Figure 1) in the Gangwon province (N38.2051°, E128.1816°). The forested headwater catchment has an area of ~16 ha, with elevations ranging from 368 to 682
5  m a.s.l. and a mean slope of 24° (Lee et al., 2016). The headwater catchment has only a narrow riparian zone around the upper part of the stream that comprises approx. 3% of the catchment area. The bedrock consists of low-permeability quarzofeldspathic orthogneiss. Soils are mostly dystric cambisols with a loamy texture and an average thickness of 0.6 m. On the hillslopes, the soil is underlain by a very hard and compact layer of hardpan-like features. A deciduous stand, resulting from natural regeneration after harvest in the 1970s, dominates at elevations above 450 m (61% of the entire area), whereas at
10  lower elevations a coniferous stand prevails that was planted after harvest at the same time (39% of the entire area). Precipitation data in daily resolution from a weather station of the Korea Meteorological Administration (station no. 594, located approx. 3 km northeast of the study site; https://www.kma.go.kr) was obtained for the years 2013 and 2014. In addition, monthly precipitation data from this station was available for the period 1997-2012. South Korea experiences the East Asian summer monsoon during the months June, July and August (JJA). Mean annual precipitation was 1273 mm (1997-2014), with
15  on average 60% of it occurring from June through August. In 2013, annual precipitation was 1313 mm (JJA: 897 mm), whereas 2014 was much drier with an annual precipitation of 699 mm (JJA: 364 mm). During the monsoon season studied in 2013, the stream surfaced 65 m upstream of the catchment outlet at low-flow conditions. During the main monsoon period in 2013, however, the stream extended 226 m upstream of the outlet, to the location of the study hillslope transect (Figure 1).

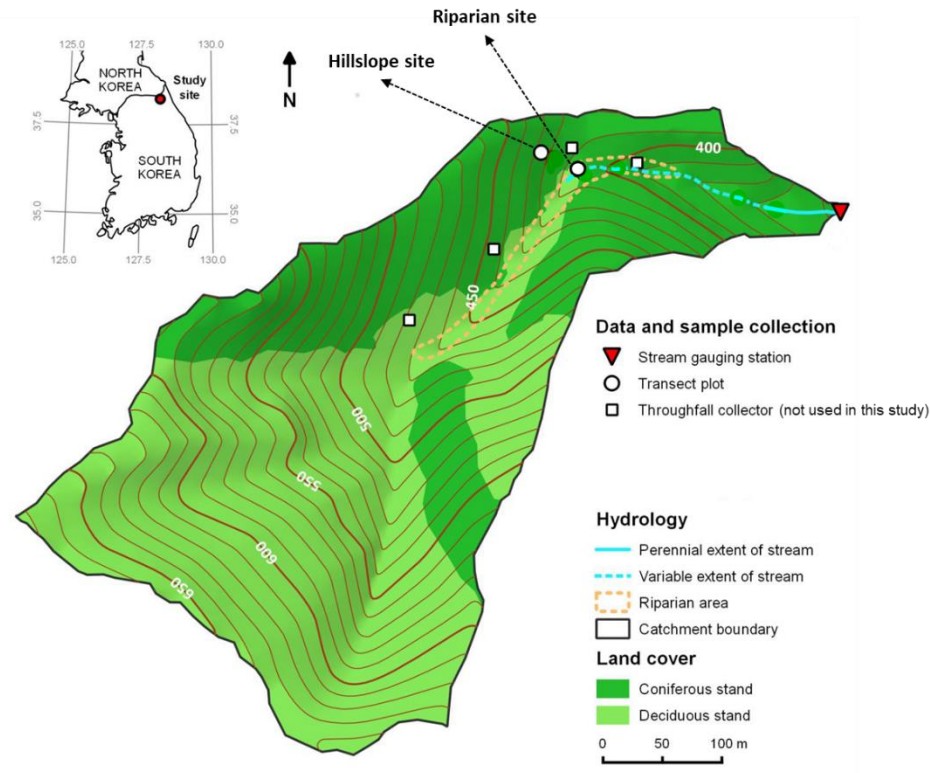

**Figure 1: Location and detailed map of the test catchment and sampling setup. Discharge was measured at a V-notch weir installed at the outlet of the catchment (red triangle).**

## 2.2 Discharge measurements

Discharge was measured at the outlet of the catchment during 2013 and 2014. Water stage was recorded at a V-notch weir every 5 min from June 1 to August 31, 2013, and from June 1 to August 16, 2014, using a pressure transducer (Levelogger Gold M10, Solinst Canada Ltd., Georgetown, Canada) that was barometrically compensated with a barometric pressure transducer (Barologger Gold M1.5, Solinst Canada Ltd., Georgetown, Canada). Discharge was calculated from stage measurements by applying a stage-discharge relationship that was developed based on the procedures outlined in WMO (2010).

Figure 2 shows daily precipitation rates and the discharge time series for the period June 9 to August 18, 2013 (corresponds to day of year DOY 160-230). This is the period for which the tracer hydrological work was performed. The monsoon season was separated into four periods, based on precipitation and hydrological response of the headwater stream. The pre-monsoon season (DOY 160-173) corresponded to baseflow conditions (49 mm of precipitation). The wet-up period (DOY 174-187) exhibited some larger rainfall events (79 mm of precipitation) that induced only a small response in discharge. The main period

(DOY 188-208) was characterized by frequent large rainfall events (564 mm total precipitation) with an increase in discharge by more than two orders of magnitude. During the drying-up period (DOY 209-230), events became infrequent again (150 mm), and discharge quickly receded.

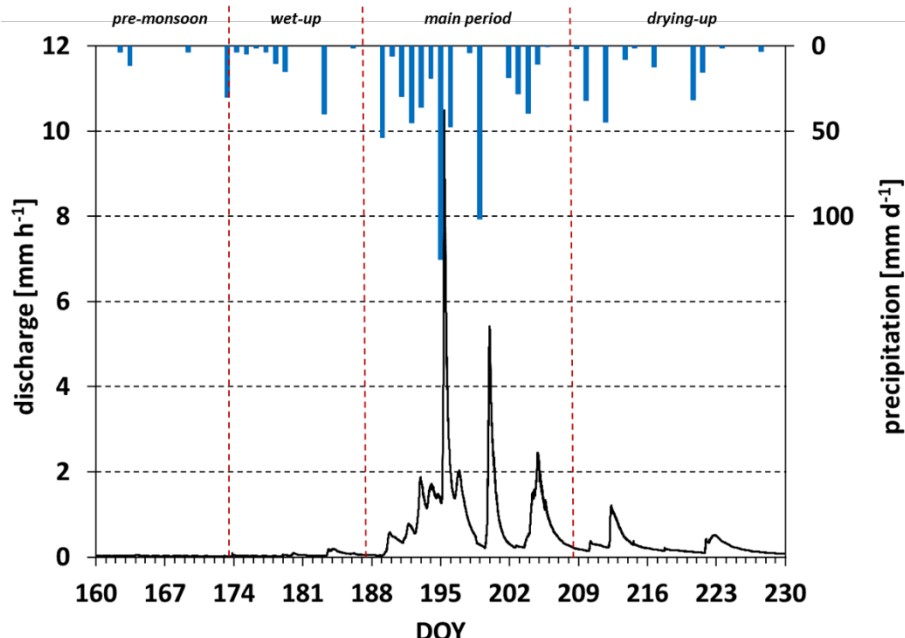

**Figure 2: Daily precipitation rates and discharge time series for the monsoon season 2013 (June 9 to August 18, i.e. day of year DOY 160-230). The monsoon season was separated into four periods, based on precipitation and hydrological response.**

## 2.3 Water sampling and chemical analyses

10    The sampling of different water sources was performed between early June and mid-August 2013. The goal was to monitor the dynamics of solute concentrations in streamflow before and during the monsoon season as well as to characterize the chemistry of soil water from different hillslope positions. Streamflow at the catchment outlet was sampled at least every two days (grab samples). During and following major rainfall events, the sampling frequency was increased to several samples per day (grab samples and automated sampling, using a 6712 Portable Sampler, Teledyne Isco Inc., Lincoln, USA).

15    Soil water was sampled every two days at two different hillslope positions, i.e. on the hillslope in a midslope position and in the riparian zone. These two positions formed a transect approx. 200 m upstream of the catchment outlet (Figure 1). Soil water was extracted using suction lysimeters, installed at 20 cm, 30 cm and 40 cm depth below the surface. Chemical analyses showed that soil water chemistry was very similar between the three depths; therefore, only values averaged across the three depths were used in this study to represent soil water from the two hillslope positions. Water samples were stored in

polypropylene test tubes at 4°C in the dark until analyses. For more detailed information on instrumentation and methodology please refer to Payeur-Poirier, 2018).

Electrical conductivity (EC) of streamflow samples and of collected soil water was measured at the time of sample collection using a portable EC meter (WTW Cond 340i, Xylem Analytics, Weilheim, Germany). Major anions and cations were also determined in the water samples but here we only report the concentrations of magnesium (Mg). Magnesium was measured by inductively coupled plasma optical emission spectrometry (Optima 3200 XL, PerkinElmer LAS GmbH, Rodgau, Germany) with a detection limit of 10 µg L$^{-1}$.

## 2.4 Deriving end-member contributions to streamflow

### 2.4.1 The hydrograph separation procedure

The procedure of hydrograph separation has the goal to separate the streamflow into its spatial or temporal components. The general procedure of hydrograph separation relies on several assumptions: (1) streamflow can be described as a linear mixture of the so-called end-members, i.e. the contributing components, (2) the end-members have characteristic and differing tracer concentrations, i.e. typical signatures, (3) end-member concentrations are time-invariant, and (4) tracers behave conservatively (Hooper et al., 1990). Any change in tracer concentration in streamflow, i.e. the mixture of components, is only due to a change in the fractional contribution of the end-members to discharge. Pairs of tracers can be explored using bivariate plots, where the concentrations of two tracers in the end-members and streamflow are plotted against each other. If streamflow can be well described by a mixture of the three selected end members, streamflow concentrations will fall within the bounds of the triangle that is created by the tracer concentrations of the three end members. Mixing ratios between the three selected end members were calculated using mass balances for water and the two tracers:

$$1 = f_1 + f_2 + f_3$$
$$c_{s1} = f_1 c_{11} + f_2 c_{21} + f_3 c_{31} \tag{1}$$
$$c_{s2} = f_1 c_{12} + f_2 c_{22} + f_3 c_{32}$$

Where $c_{sj}$ means concentration of tracer $j$ in streamwater, $c_{ij}$ is concentration of tracer $j$ in end member $i$, and $f_i$ is fractional contribution of end member $i$ to streamflow. By rearranging these three equations, the three unknowns $f_1$, $f_2$ and $f_3$ can be determined.

### 2.4.2 Tracer time series in streamflow

For this study, EC and Mg were used as tracers. During the pre-monsoon and the wet-up period, tracer values in streamflow remained relatively stable (Figure 3). With the onset of the main period, however, tracer values decreased markedly. Towards the end of the drying-up period, EC values and Mg concentrations started to increase again.

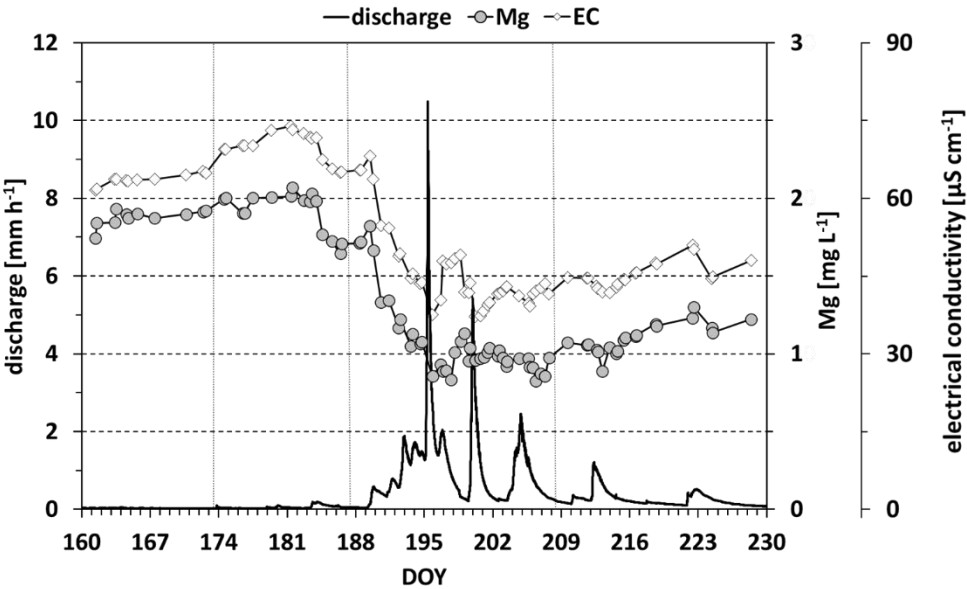

**Figure 3: Time series of discharge and of EC and Mg in streamflow. Vertical dashed grey lines separate the four monsoon periods (see also Figure 2).**

### 2.4.3 Characterizing the tracer signature of the end-members

We defined three end-members, i.e. three water sources potentially contributing to streamflow: hillslope soil water, riparian zone soil water, and groundwater. During the pre-monsoon season, i.e. baseflow conditions, we assumed groundwater to be the only component contributing to streamflow. Since we did not sample groundwater directly, we used the average of the EC values and Mg concentrations measured in streamflow during the pre-monsoon period (DOY 160-173) as the tracer signature of the groundwater end-member. As overland flow was not observed during the field work in 2013 and also direct channel interception was assumed to be negligible in this headwater catchment, we did not consider throughfall to directly contribute to streamflow and, therefore, did not include it in the hydrograph separation. Hillslope soil water and riparian zone soil water, sampled as described above, were assumed to contribute via subsurface flow pathways to streamflow.

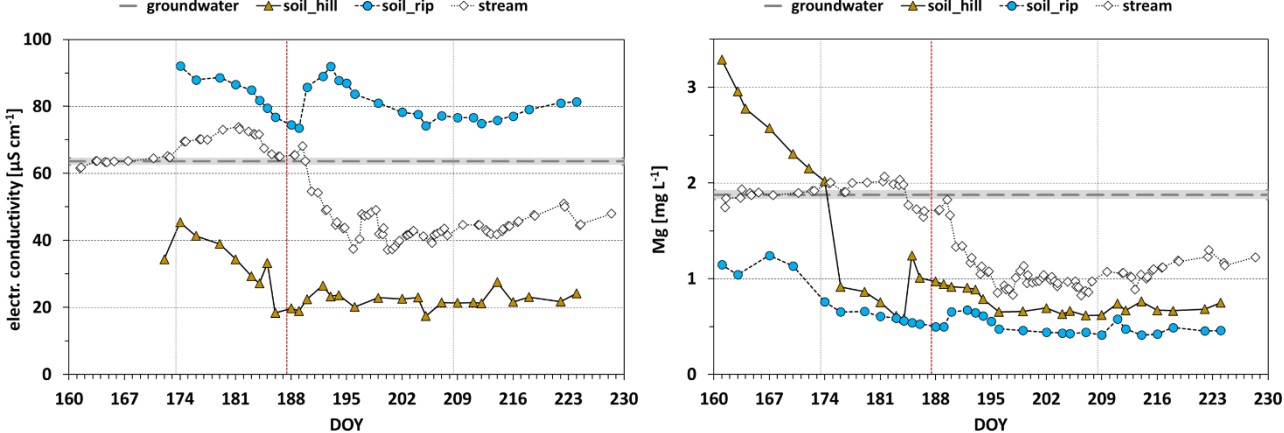

**Figure 4: Time series of electrical conductivity (left) and magnesium (right) in streamflow and in the end-members hillslope soil water (soil_hill) and riparian zone soil water (soil_rip). For the groundwater end-member, the mean of baseflow concentrations during the pre-monsoon period (DOY 160-173) is shown, as well as the standard deviation (n=11) as grey band. Vertical dashed grey lines separate the four monsoon periods (see also Figure 2). The red dashed line signifies the onset of the main period of the monsoon.**

Hillslope soil water as well as riparian zone soil water showed strongly varying tracer values during the pre-monsoon and wet-up period, i.e. before the onset of the main period, thereby violating the assumption (3) for hydrograph separation listed above (Figure 4). From DOY 188 on, however, EC values and Mg concentrations remained fairly stable in hillslope soil water (coefficients of variation 11% and 16%, respectively) as well as riparian zone soil water (coefficients of variation 7% and 17%, respectively). Therefore, mean end-member tracer signatures were only calculated for the period DOY 188-230, and the three-component hydrograph separation was only performed for this period, i.e. for the main period and the drying-up. Based on EC values and Mg concentrations in the stream (Figure 4) and also general streamwater chemistry, we concluded that from DOY 160 to DOY 187, i.e. also during the wet-up period, streamflow was primarily composed of groundwater. In contrast, streamflow tracer values during the main and drying-up period could well be described by a linear combination of the three selected components (Figure 5).

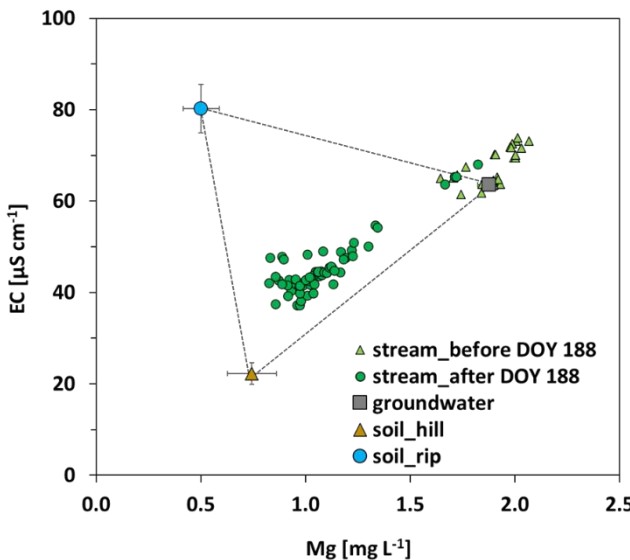

**Figure 5: Mixing diagram showing streamflow tracer values for EC and Mg, separated in the period before and after DOY 188 (i.e. onset of main period), and end-member tracer signatures, including standard deviations (calculated for DOY 188-230).**

5 **2.4.4 A switch in end-member contributions during the main monsoon period**

The hydrograph separation results indicated that during the main period and the drying-up, the groundwater contribution decreased considerably, and the signatures of hillslope soil water and riparian zone soil water became discernible in streamflow, suggesting a substantial contribution to streamflow from the hillsides of the catchment.

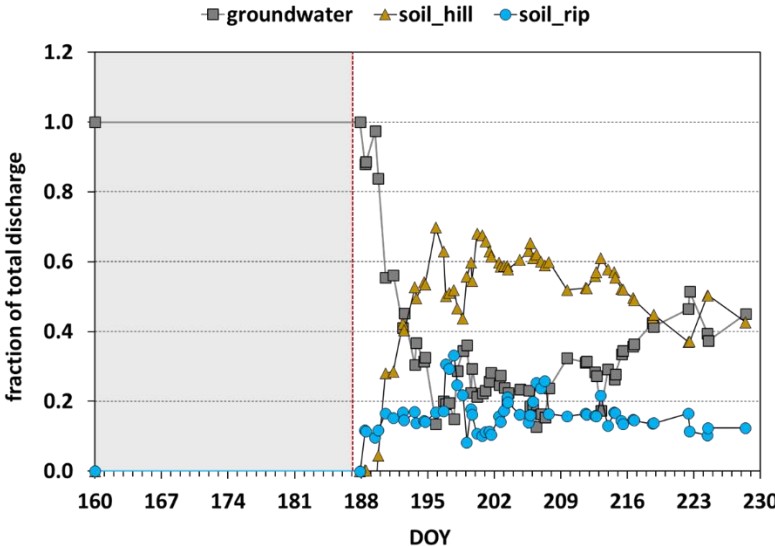

**Figure 6: Contributions of the three selected end-members to total discharge for the period DOY 188-230. The red dashed line indicates the onset of the main period of the monsoon. Prior to DOY 188, streamflow was primarily composed of groundwater.**

The contribution of groundwater to streamflow dropped from 100% to values between 20% and 40% (mean 34%) during the main and drying-up period (Figure 6). The contribution from riparian zone soil water varied mostly between 10% and 21% (mean 16%), whereas hillslope soil water contributed between 40% and 60% (mean 50%). This indicates that hydrological connectivity between the hillslopes and the stream was established, and the chemical composition of streamflow was dominated by the hillslope soil water signatures for the main and drying-up periods. This observation is in contrast to other

studies that have emphasized the dominant role of the riparian zone in controlling the chemistry of the subsurface flow that enters the stream (Klaus and Jackson, 2018; Ledesma et al., 2018; Bishop et al., 2004; Cirmo and McDonnell, 1997). The specific topography of our test catchment with steep hillslopes and narrow riparian zones in combination with heavy rainfall events during the intense phase of the monsoon season results in hillslope-generated subsurface stormflow passing through the riparian zone without undergoing mixing processes. Therefore, the hillslope soil water signature can be detected in streamflow.

Most likely, this direct hillslope soil water contribution to streamflow will subside once the headwater catchment drains and discharge returns to baseflow conditions.

### 3 Methods

We used a process-based lumped model to simulate the storage and flow dynamics of the hillslope, the riparian zone and the

groundwater for different periods of the 2013 monsoon season by separate subroutines. We used a Monte Carlo approach to

create 2,000,000 simulation time series, which we iteratively confined using performance criteria of discharge and the mixing ratios estimated by tracer-based three component hydrograph separation (Table 1). At each step, we quantify the identifiability of model parameters to learn about the usefulness of the discharge observations and hydrograph separation results considered in the confinement procedure. We finally compare the uncertainty of the simulated streamflow components with and without using the hydrograph separation results and, using independent discharge observations of the 2014 monsoon season, quantify how much the inclusion of experimentally derived streamflow components can reduce prediction uncertainty.

## 3.1 The model

We use a modified version of the HBV model (Beck et al., 2010; Seibert and Vis, 2012). The model was modified to include the riparian zone similar to (Seibert et al., 2003) and simplified by removing the snow routine and considering only two reservoirs that simulate the contributions of the hillslope, the riparian zone and groundwater to total discharge with eight model parameters (Figure 7, Table 1). The soil storage receives all precipitation [mm/d] and calculates actual evapotranspiration [mm/d] from potential evaporation [mm/d] (Penman-Wendling approach, (DVWK, 1996; Wendling et al., 1991) by multiplication with an evaporation factor $f_{Evap}$ [-] ($0 \leq f_{Evap} \leq 1$):

$$f_{Evap}(t) = \frac{V_S(t)}{F_C \cdot L_P} \tag{2}$$

with $V_S$ [mm] as the soil storage at time $t$, $F_C$ [mm] the field capacity, and $L_P$ [-] as an evaporation shape factor. A wetness factor $f_{Wet}$ derived from soil saturation and a shape factor $\beta$ [-] determines the fraction of precipitation that percolates through the soil:

$$f_{Wet}(t) = \left(\frac{V_S(t)}{F_C}\right)^{\beta} \tag{3}$$

The remaining part of precipitation [$1 - f_{wet}(t)$] is added to the soil storage. Soil percolation is added to the water stored in reservoir one, $V_1(t)$ [mm], which is drained by groundwater discharge $Q_{GW}$ [mm/d] and hillslope discharge (sometimes referred to by subsurface storm flow or interflow) $Q_{HS}$ [mm/d] when a maximum groundwater storage $U_{GW}$ [mm] is exceeded. This model process represents conceptually the impact of rising groundwater levels on lateral transmissivities that allow fast saturated flow down the hillslope towards the riparian zone.

$$Q_{GW}(t) = \frac{V_1(t)}{K_{GW}} \tag{4}$$

$$Q_{HS}(t) = \begin{cases} \frac{V_1(t) - U_{GW}}{K_{HS}} & if \ V_1(t) \geq U_{GW} \\ 0 & if \ V_1(t) < U_{GW} \end{cases} \tag{5}$$

where $K_{GW}$ [d] and $K_{HS}$ [d] are the storage constant of the groundwater and the hillslope, respectively, and $U_{GW}$ [mm] is the maximum groundwater storage. Hillslope discharge is fed into reservoir two, which represents the riparian zone until riparian zone storage $V_2(t)$ exceeds it maximum capacity $U_{RZ}$ [mm]. Discharge of the riparian is therefore defined as

$$Q_{RZ}(t) = \begin{cases} \dfrac{V_2(t)}{K_{RZ}} & \text{if } V_2(t) < U_{RZ} \\ \dfrac{U_{RZ}}{K_{RZ}} & \text{if } V_2(t) = U_{RZ} \end{cases} \qquad (6)$$

5  Catchment discharge is obtained by summarizing over $Q_{GW}$, $Q_{HS}$ and $Q_{RZ}$ at each time $t$ and rescaling them to [m³/s] using the catchment area (16 ha). Re-scaling the catchment discharge for each time step $t$, we can express each streamflow component in [%]. Similar to preceding work that compared simulated and tracer derived streamflow contributions (Robson et al., 1991, 1992), we can now compare the model's simulations to the results of the streamflow separation analysis (section 2).

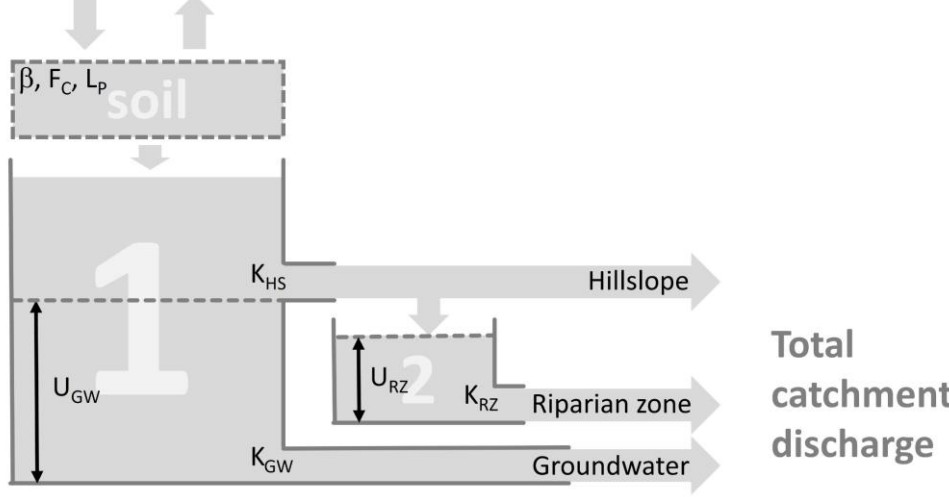

**Figure 7: Structure of the modified HBV model. The three components hillslope, riparian zone and groundwater sum up to total catchment discharge.**

The model operates at a daily temporal resolution to simulate the monsoon seasons of 2013 and 2014 after a warm-up period of 3.5 years. Precipitation data from a nearby meteorological station of the Korean Meteorological Administration (see section

15  2) and from a global product (Global Land Data Assimilation System GLDAS,(Rodell et al., 2004), corrected with the observations from the local weather station, were used to complete the missing observations before the 2013 monsoon season and between the two monsoon seasons. Since reliable hydrograph separation results are only available for the 2013 monsoon season, we use this year for model calibration, whereas the monsoon season of 2014, for which only discharge observations are available, was used for the validation of the model.

## 3.2 Step-wise parameter estimation and quantification of parameter identifiability

Similar to the Generalized Likelihood Uncertainty Estimation (GLUE) framework (Beven and Binley, 1992), we use a "soft rules" approach to estimate model parameters and their identifiability that allows the consideration of different types of observations (Hartmann et al., 2017; Sarrazin et al., 2018; Chang et al., 2020). We apply a Monte Carlo parameter sampling to obtain 2,000,000 model realisations derived by uniform sampling of model parameters within their predefined ranges (Table 1). For each run, we calculate the model performance concerning observed catchment discharge by the Kling Gupta efficiency $KGE_Q$ (Gupta et al., 2009), that indicates flawless simulations with a value of one and simulations worse than the simple average of the observations with a value of -0.41 (Knoben et al., 2019), and the deviation of observed and simulation contributions of groundwater $F_{GW}$ [%] and midslope discharge $F_{HS}$ [%] over the two monsoon sub-periods, for which stable end-member estimates were available (pre-monsoon and wet-up, and main monsoon and drying up, as defined in subsection 2.4)**Fehler! Verweisquelle konnte nicht gefunden werden.**. In a three-step procedure, we remove those model realisations that perform poorly against discharge or streamflow contribution observations with rather soft thresholds for $F_{GW}$ and $F_{HS}$ to account for the comparably large uncertainties of multi-component streamflow separation (Genereux, 1998) and simplifications of our simulation model (see subsection 3.1).

1. We reduce the sample by discarding all simulations that perform badly in terms of observed total streamflow by removing all simulations with $KGE_Q < 0.8$.

2. We further reduce the sample by removing all simulations whose $F_{HS}$ show more than 10% deviation from the hydrograph separation estimates. The relatively large value of 10% was chosen because of the uncertainty of the end-members (as described in subsection 2.4) and previous hydrograph separations (Genereux, 1998). It is final value of 10% found by a trial-and-error-procedure, which accounts also for the uncertainties arising simplifications in our simulation model.

3. We further reduce the sample by removing all simulations whose $F_{GW}$ show more than 20% deviation from the hydrograph separation estimates. Since the contributions of the hillslope, groundwater and the riparian zone sum up to 100%, riparian zone contributions are implicitly considered in this last step.

To estimate changes of identifiability of the model parameters through adding more and more information along the three parameter confinement steps, we quantify the strength of reduction of the initial sample of 2,000,000 and the change of the distribution of each model parameter at each individual step. If discharge observations or one of the hydrograph separation streamflow components provides information to better estimate model parameters, a strong decrease of the initial sample and a substantial change of a large number of model parameters should be found. To analyse the sensitivity of our results to the selection of the two thresholds ($KGE_Q < 0.8$, and $F_{HS}$ and $F_{GW} \pm 10\%$), we relax their values and repeat the analysis two times. Once with $KGE_Q < 0.5$ (and $F_{HS}$ and $F_{GW} \pm 10\%$), and $F_{HS}$ and $F_{GW} \pm 20\%$ (and $KGE_Q < 0.8$).

### 3.3 Quantification of uncertainty of simulated model internal fluxes and discharge

We quantify simulation uncertainty of discharge by the mean and standard deviations of $KGE_Q$, obtained by using only observed discharge or both observed discharge and the hydrograph separation results for parameter confinement for the calibration period in 2013 and the validation period in 2014. Similarly, to quantify the simulation uncertainty of simulated internal fluxes (hillslope discharge, groundwater discharge and riparian zone discharge), we compare their simulated means and standard deviations, that were obtained by using only observed discharge or by both observed discharge and the hydrograph separation results for parameter confinement, with the hydrograph separation derived streamflow components during the two time periods of the 2013 sampling period. We do the same for the 2014 monsoon season but since there are no reliable hydrograph separation results available for this year, we only analyse the simulated mean and standard deviation of the simulated streamflow contributions for both calibrations. If the hydrograph separation derived from streamflow components provides new information for parameter estimation, it will result in a reduction of uncertainty of simulated fluxes and discharges in both years, and an increase of $KGE_Q$ of the 2014 predictions should be found. In order to better interpret model performances and simulation uncertainties, we calculate additional performance metrics (equations provided in supplemental information SI): the Nash-Sutcliffe efficiencies, the logarithmic Nash-Sutcliffe efficiency, the Root Mean Squared Error, and the individual components of the Kling Gupta efficiency (bias, variability and correlation).

### 4 Results

### 4.1 Step-wise parameter estimation and quantification of parameter identifiability

When iteratively applying the three rules for parameter confinement, we observe a substantial decrease of the initial sample of 2,000,000 parameter sets (Figure 8). Extracting only those with $KGE_Q \geq 0.8$ reduces the sample to less than 10% (137,137 parameter sets left). Adding the observed streamflow components to the calibration procedure results in a further reduction of the sample. Discarding all parameter sets that deviate more than 10% from the observed hillslope contributions, results in 2,786 remaining parameter sets and in 56 parameter sets when the groundwater contributions (implicitly the riparian zone contributions, too) are finally added. Despite being only average values over the two sub-periods of the 2013 sampling period, the incorporation of the hydrograph separation derived streamflow contributions results in a reduction by more than three orders of magnitude, while the discharge observations, although using a high value of 0.8 of the $KGE_Q$ criterion, only reduced the sample by slightly less than one order of magnitude.

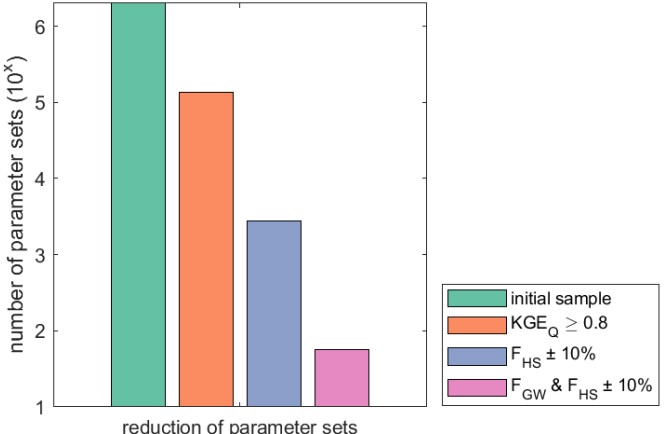

**Figure 8: Iterative reduction of the initial sample of 2,00,000 parameter sets using the KGE$_Q$ and hydrograph-separation derived streamflow contributions for the individual years 2013 and 2014, as well as for both years together**

The influence of the parameter confinement procedure using observed discharge and streamflow components is also visible through the changes of the distribution of each of the parameters occurring at each of the confinement steps (Figure 9). When only discharge is considered in the first step of the confinements (KGE$_Q \geq 0.8$), some model parameter distributions shift away from the mean of the normalized range, e.g. $L_P$, $K_{HS}$, or $K_{GW}$, but only one of them, $F_C$, shows a confinement of its 25$^{th}$ and 75$^{th}$ percentile, which indicates a reduction of uncertainty. When the sample is further confined by the observed streamflow contributions of the hillslope, a few more parameters shift away from the mean, e.g. $U_{GW}$ and $U_{RZ}$, but three more parameters, $K_{HS}$ and $K_{RZ}$, show confined uncertainties. When finally adding the groundwater contributions (and implicitly the riparian zone contributions), almost all model parameters show a clear shift of their distributions away from the mean, for most of them going along with a reduced uncertainty indicated by narrowing 25$^{th}$ and 75$^{th}$ percentiles. We find the same results when calculating the mean and standard deviations of the model parameters for the confinement by discharge only and the confinement by discharge and the experimentally derived (i.e., tracer-based) contributions to streamflow (Table 1).

**Table 1: Parameters of the modified HBV model, description, units, and boundaries for parameter estimation (see below), and the model performances and simulated streamflow components for the two delineated monsoon periods (see section 2) when confining the initial parameter sample by discharge only, and by discharge and tracer-based contributions to streamflow for the calibration in 2013 and the validation in 2014**

| Parameter | Description | Unit | Lower boundary | Upper boundary | KGE$_Q \geq 0.8$ | F$_{GW}$ & F$_{HS}$ ± 20% |
|---|---|---|---|---|---|---|
| $\beta$ | Shape factor | [-] | 1 | 10 | 5.2 ± 2.6 | 5.1 ± 2.5 |
| $F_C$ | Maximum storage in hillslope soil | [mm] | 0 | 250 | 119.3 ± 66.8 | 61.4 ± 64.7 |
| $L_P$ | Threshold for reduction of evaporation | [-] | 0.3 | 1 | 0.6 ± 0.2 | 0.6 ± 0.2 |
| $log10\ K_{HS}$ | Recession coefficient (hillslope) | [d$^{-1}$] | -5 | 0 | -1.6 ± 1.3 | -1.1 ± 0.4 |
| $log10\ K_{GW}$ | Recession coefficient (groundwater) | [d$^{-1}$] | -5 | 0 | -1.7 ± 1.5 | -1.6 ± 0.4 |
| $U_{GW}$ | Maximum groundwater storage | [mm] | 0 | 250 | 126 ± 70.4 | 32.2 ± 30.9 |
| $log10\ K_{RZ}$ | Recession coefficient (riparian zone) | [d$^{-1}$] | -5 | 0 | -2.5 ± 1.5 | -4.1 ± 0.7 |

| | | | | | | | |
|---|---|---|---|---|---|---|---|
| | $U_{RZ}$ | Maximum riparian zone storage) | [mm] | 0 | 100 | 52.8 ± 30.2 | 48.4 ± 29.5 |
| **2013** | KGE$_Q$ | Kling-Gupta efficiency concerning discharge | [-] | ∞ | 1 | 0.84 ± 0.02 | 0.84 ± 0.02 |
| | NSE$_Q$ | Nash-Sutcliffe efficiency concerning discharge | [-] | ∞ | 1 | 0.76 ± 0.04 | 0.75 ± 0.04 |
| | logNSE$_Q$ | Nash-Sutcliffe efficiency concerning log-discharge | [-] | ∞ | 2 | -0.51 ± 0.05 | -0.49 ± 0.05 |
| | RMSE | Root mean squared error concerning discharge | [l³s⁻¹] | 0 | ∞ | 12.03 ± 0.89 | 12.27 ± 0.91 |
| | β$_Q$ | bias of the simulated and observed discharges | [-] | 0 | ∞ | 0.93 ± 0.04 | 0.93 ± 0.04 |
| | α$_Q$ | relative variability in the simulated and observed discharges | [-] | 0 | ∞ | 0.94 ± 0.04 | 0.94 ± 0.04 |
| | r$_Q$ | linear correlation between simulated and observed discharges | [-] | -1 | 1 | 0.88 ± 0.01 | 0.88 ± 0.01 |
| | F$_{GW,BF}$ | simulated groundwater contribution during pre-monsoon | [-] | 0 | 1 | 0.74 ± 0.32 | 0.99 ± 0.02 |
| | F$_{GW,,MM}$ | simulated groundwater contribution during main monsoon | [-] | 0 | 1 | 0.6 ± 0.46 | 0.44 ± 0.02 |
| | F$_{MS,BF}$ | simulated midslope contribution during pre-monsoon | [-] | 0 | 1 | 0.06 ± 0.16 | 0 ± 0 |
| | F$_{MS,MM}$ | simulated midslope contribution during main monsoon | [-] | 0 | 1 | 0.3 ± 0.4 | 0.45 ± 0.04 |
| | F$_{RZ,BF}$ | simulated riparian contribution during pre-monsoon | [-] | 0 | 1 | 0.2 ± 0.27 | 0.01 ± 0.02 |
| | F$_{RZ,MM}$ | simulated riparian contribution during main monsoon | [-] | 0 | 1 | 0.1 ± 0.23 | 0.12 ± 0.05 |
| **2014** | KGE$_Q$ | model performance concerning discharge | [-] | ∞ | 1 | -0.98 ± 1.54 | -0.02 ± 0.34 |
| | NSE$_Q$ | Nash-Sutcliffe efficiency concerning discharge | [-] | ∞ | 1 | -7.47 ± 14.73 | -1.75 ± 1.31 |
| | logNSE$_Q$ | Nash-Sutcliffe efficiency concerning log-discharge | [-] | ∞ | 2 | 0.14 ± 0.08 | 0.08 ± 0.04 |
| | RMSE | Root mean squared error concerning discharge | [l³s⁻¹] | 0 | ∞ | 1.22 ± 0.76 | 0.8 ± 0.19 |
| | β$_Q$ | bias of the simulated and observed discharges | [-] | 0 | ∞ | 0.33 ± 0.32 | 0.444 ± 0.32 |
| | α$_Q$ | relative variability in the simulated and observed discharges | [-] | 0 | ∞ | 0.61 ± 1.69 | 0.38 ± 0.41 |
| | r$_Q$ | linear correlation between simulated and observed discharges | [-] | -1 | 1 | 0.51 ± 0.24 | 0.56 ± 0.05 |
| | F$_{GW,MS}$ | simulated groundwater contribution monsoon season* | [-] | 0 | 1 | 0.68 ± 0.31 | 0.88 ± 0.1 |
| | F$_{MS,MS}$ | simulated midslope contribution during monsoon season* | [-] | 0 | 1 | 0.07 ± 0.13 | 0.02 ± 0.01 |
| | F$_{RZ,MS}$ | simulated riparian contribution monsoon season* | [-] | 0 | 1 | 0.25 ± 0.27 | 0.1 ± 0.1 |

* from 01.04.2014 to 30.09.2014

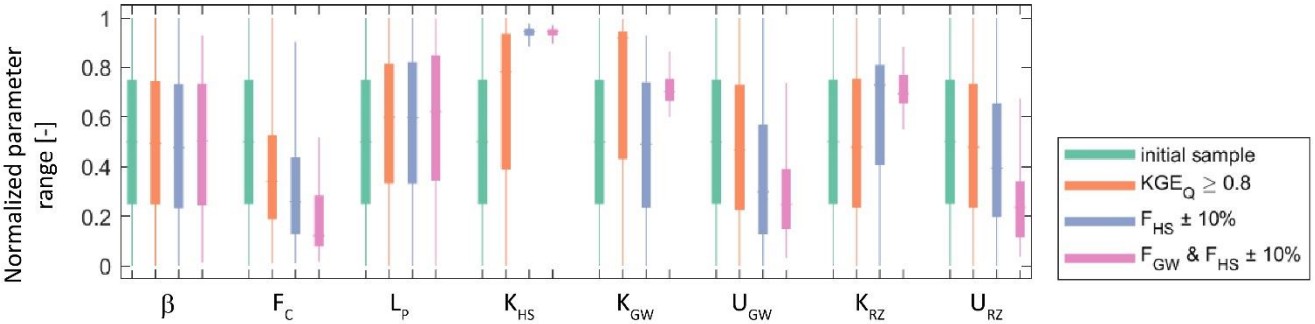

**Figure 9: Initial parameter distribution and their modification along the three parameter estimation steps for the individual years 2013 and 2014, as well as for both years together. Boxes indicate the range between the 25th and 75th percentile, lower and upper whiskers show the 5th and 95th percentile, respectively.**

Changing the thresholds towards more relaxed rules, once with KGE$_Q$ < 0.5 (and $F_{HS}$ and $F_{GW}$ ± 10%), and $F_{HS}$ and $F_{GW}$ ± 20% (and KGE$_Q$ < 0.8), results in a weaker reduction of the initial sample of 2,000,000 parameter sets, most pronounced when relaxing the criteria for the streamflow components towards $F_{HS}$ and $F_{GW}$ ± 20% (Figure S1 of the SI). Consequently, weaker confinements of the parameter distributions are found whereas $K_{HS}$, $K_{GW}$, $K_{RZ}$, and $U_{RZ}$ seem to remain identifiable despite relaxing KGE$_Q$, while $F_C$, $K_{HS}$, $K_{GW}$, and $U_{GW}$ seem to unaffected by the relaxing of $F_{HS}$ and $F_{GW}$ ((Figure S2 of the SI).

**4.2 Quantification of uncertainty of simulated model internal fluxes and discharge**

Using only $KGE_Q \geq 0.8$ to confine the parameter sample, an average $KGE_Q$ of 0.839 with a relatively low standard deviation of 0.024 is found for the calibration period in 2013 (Table 1), which also results in an acceptable visual agreement between simulations and observations (Figure 10g). Adding the experimentally derived contributions to streamflow to the parameter

confinement results in almost the same mean $KGE_Q$ (0.840), standard deviation (0.023), and visual agreement. However, when looking at the simulated streamflow contributions of the calibration by discharge only, we find that the standard deviations are large compared to the mean simulated contributions of groundwater, hillslope discharge and riparian zone discharge across all two monsoon periods (Table 1). Visualizing the entire range of their uncertainties (Figure 10a,c,e), we can see that simulated groundwater and riparian zone contribution could range from 0% to 100%. The same is true for the hillslope contributions

during wet-up, main monsoon and drying up. Only during drier periods, hillslope contributions to discharge are limited and sometimes fall down to 0%. Adding the experimentally derived contributions to streamflow  to the parameter confinement reduces the simulation uncertainty of all three streamflow components for the two monsoon periods in 2013 as indicated by their strongly reduced standard deviations in Table 1 and by the narrower ranges around the observations of their simulations in Figure 10a,c,e. The strong dominance of the groundwater streamflow component during the baseflow and wet-up periods is

well represented, as well as the onset of hillslope discharge during the main monsoon and the drying period, when the contributions of the riparian zone to streamflow gradually increase. The simulations also indicate that hillslope discharge mostly replaces groundwater in the main monsoon and the drying up, while before and after the monsoons season, streamflow is comprised by an interplay of groundwater and riparian zone discharge. The comparison of simulations and observations also indicates that strong variations of streamflow components occur even within the monsoon periods, especially during the main

monsoon and the drying-up (Figure 10a,c,e).

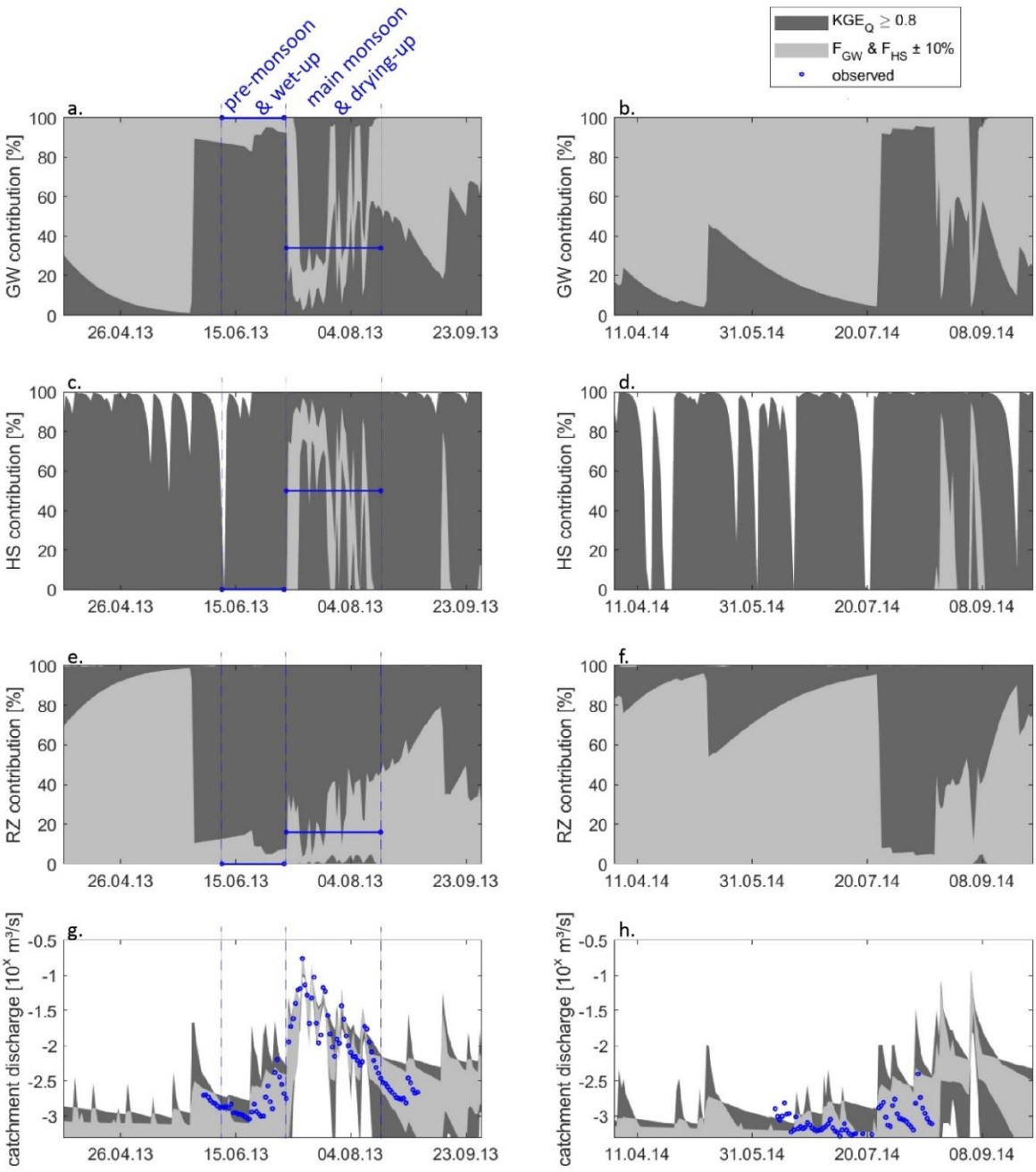

**Figure 10: Simulated time series of contributions of groundwater (a and b), subsurface stormflow (c and d), and riparian zone discharge (e and f), total catchment discharge (g and h; blue points represent discharge observations from the test catchment, blue lines indicate the experimentally derived contributions to streamflow, averaged over pre-monsoon and main monsoon, see subsection 2.4 ) by using discharge (KGE_Q) only and by using F_SSF and F_GW during both years for parameter estimation.**

During the validation in 2014, simulation performance of discharge decreases for both calibration steps (Table 1). Using only discharge observations, a very poor simulation quality (KGEQ=-0.98) is found with a standard deviation of 1.54 indicating a very high simulation uncertainty. When using both, discharge and streamflow components for calibration, a much better performance is found (KGEQ=0.02), which is well above the KGE that would be obtained when using just the average

observations to predict discharge (-0.41) and which has a much smaller simulation uncertainty indicated by a standard deviation of 0.09, which is confirmed when comparing simulated and observed time series (Figure 10h). Although there are no observations of the streamflow components available for the validation year 2014, we can still see that the simulation uncertainty of all three components indicated by their standard deviations is generally high over the whole simulation period when only discharge is used for calibration and reduced by more than a third when the stream contributions are considered in

the calibration (Table 1). Similar to the calibration year 2013, we see that the interplay between groundwater and the riparian zone is much better defined and that the short but pronounced initiation of hillslope discharge is much better represented when both observed discharge and stream flow components are used for calibrations (Figure 10b,d,f).

Considering the other discharge performance metrics, we see that $NSE_Q$ and the individual components of the KGE ($\beta_Q$, $\alpha_Q$, and $r_Q$) reflect what is already shown by $KGE_Q$, i.e. a high simulation performance of discharge for both calibration types for

the year 2013. Likewise, $RMSE_Q$ indicates small errors in the range of ~0.012 m³/s. Only $logNSE_Q$ deviates from the general impression of acceptable model performance indicating poor simulation performance of the model for low flows for both calibration types. For 2014, we find that $NSE_Q$ and the individual components of the KGE ($\beta_Q$, $\alpha_Q$, and $r_Q$) again reflect what is shown by $KGE_Q$, in this case inferior performance compared to 2013. But $logNSE_Q$ and $RMSE_Q$ indicate a lower simulation error and a better low flow performance for 2014, respectively.

**5 Discussion**

**5.1 Realism of model simulations**

We use a simple approach to incorporate streamflow contributions derived from environmental tracers into our simulation approach that compares simulated streamflow contributions and tracer-derived streamflow contributions instead of simulating tracer transport directly. That way, no additional uncertainty due to additional model parameters was introduced due to

additional model parameters to consider transport (Birkel and Soulsby, 2015). Despite its simple structure, the model easily achieves performances of KGE ≥ 0.8 with more than 130,000 parameter sets (out of initially 2,000,000, Figure 7) indicating adequateness of its structure for simulating the hydrology of our small forested mountainous catchment. Such good performance could be expected since similar models as the HBV model or similar modifications of the HBV model performed already well at similar landscapes (Seibert et al., 2003; Uhlenbrook et al., 1999; Chen et al., 2018). Including the experimentally

derived contributions to streamflow results in a further substantial reduction of the initial parameter sample to 56 parameter sets and in a slight decrease of overall discharge simulation performance as indicated by different performance metrics (Table 1). Such further reduction of the parameter sample is due to the increased difficulty to simulate adequately and simultaneously

both discharge and streamflow contributions and was already found in previous studies that investigated the influence of additional information in a GLUE-like approach (Mudarra et al., 2019; Hartmann et al., 2017). Relaxing the thresholds to confined the sample, resulted in weaker reductions of the initial parameter sample and parameter distributions that indicate lower parameter identifiability but the overall results of the step-wise parameter estimation did not change (Figures S1,S2 of the SI).

Likewise, a decrease of simulation uncertainty concerning discharge going along with incorporating additional information into parameter estimation has already been observed (Birkel et al., 2014; Yang et al., 2021; Seibert and McDonnell, 2002). This mostly went along with an increased identifiability of model parameters and prediction skill, which is also found in this study. Using only discharge for model parametrization, a mean KGE of -0.98 in the validation year of 2014 is found (Table 1). The parameter sets obtained from using both discharge and experimentally derived contributions to streamflow result in a mean KGE of 0.02. Compared to the performances of KGE $\geq$ 0.8 that we obtained during the calibration in year 2013, this appears to be a strong decrease but it is substantially better than using the mean of discharge observations for prediction (that would result in KGE=-0.41; (Knoben et al., 2019). Also, while the discharge observations in the calibration year 2013 cover the entire stream response to the monsoon season (maximum observed discharge > 0.15 m³/s, Figure 10), the validation time period of 2014 it is much dryer than 2013 and it stops before the onset of the late and weak monsoon events in late August that produced increased discharge observations (observed discharges < 0.004 m³/s, Figure 10). For that reason, we consider the evaluation more rigorous through the challenge of predicting low flows with a calibration period of 2013 covering the entire variability of streamflow (Nicolle et al., 2014). This is also supported by the $RMSE_Q$ and $logNSE_Q$ values of the discharge predictions that indicate a lower simulation error and a better low flow performance for 2014, respectively.

**5.2 Identification of model parameters and processes**

The acceptable multi-variate performance of the model in the calibration period and the still acceptable performance found in the validation period gives us reason to believe that our approach provides interpretable results. Incorporating experimentally derived contributions to streamflow into parameter estimation results in reduced parameter uncertainty for all model parameters except for $\beta$ and $L_P$ (remain the same) compared to the parameter estimation using discharge only (Table 1). The iterative inclusion of observations into the parameter estimation procedure allows assessing the usefulness of each type of information. When discharge only is considered, changes of the distributions of parameters $L_P$, $K_{HS}$, or $K_{GW}$, and $F_C$ occur (Figure 9), confirming the well-known fact that only four to six model parameters can be identified when calibrating a model with discharge observations only (Ye et al., 1997; Wheater et al., 1986; Jakeman and Hornberger, 1993). When the experimental information of the contributions of the hillslope subsurface flow to streamflow is added, more parameters change their distributions indicating that additional information is added to the parameter estimation. We can see that this is most pronounced for $K_{HS}$, which controls the discharge dynamics of the hillslope, and $U_{GW}$ that indirectly controls hillslope discharge by triggering it after saturation of the groundwater storage (Figure 7). Adding the experimentally derived contributions of groundwater (and implicitly information about the riparian zone contributions as all three together sum up to 1), we see an

increase of identifiability for $K_{GW}$, which is indicated by a further narrowing of its 25$^{th}$ and 75$^{th}$ percentile. Most prominently, $K_{RZ}$ and $U_{RZ}$ show substantial confinement indicating the new information about streamflow contributions added more information about riparian zone and groundwater dynamics.

Previous work with a model that simulated discharge and solute transport already showed that added information through environmental tracers can be linked to their origin in the hydrological system and respective model parameters (Yang et al., 2021; Hartmann et al., 2017; Birkel et al., 2014). Our results indicate that even without the explicit inclusion of solute transport in the model, similar linkages between observations of streamflow contributions and model parameters that control the dynamics of their origin, hillslope, groundwater or riparian zone, could be found. These relationships are plausible and can be regarded as validation of the realism of the model structure (McMillan et al., 2012; Capell et al., 2012; Hartmann et al., 2013). By including discharge and observed streamflow components into parameter estimation without adding more complexity to the model, we achieve desirable levels of model parameter identifiability (eight out of nine parameters) and prediction uncertainty (Birkel and Soulsby, 2015). The resulting parameters express the effective properties of our test catchment with thin soil ($F_C$ = 61.4 mm ± 64.7 mm) and fast percolation of water towards the hillslope and groundwater storages through a high value of $\beta$ (5.1 ± 2.5). The value of $L_P$ (0.6 ± 0.2) indicates that plant water uptake through forest cover is efficient even below saturation of the soil. The groundwater storage can store more than double of the soil while the riparian zone storage is about 15 mm smaller (Table 1). With around 0.08 d$^{-1}$, $K_{HS}$ indicates fast hillslope dynamics after initiation, while at around 0.025 d$^{-1}$ and below, $K_{GW}$ and $K_{RZ}$ are reacting slowly. The scales of the three parameters are comparable to the parameters identified by (Uhlenbrook et al., 1998), who found 0.1 – 0.35 d$^{-1}$ and 0.02 – 0.05 d$^{-1}$ for their simulated interflow and groundwater dynamics, respectively.

**5.3 Benefits of including experimentally derived contributions to streamflow for streamflow prediction**

The simulated streamflow contributions obtained by discharge during the same period show considerable uncertainty allowing for contributions of groundwater and the riparian zone from 0% - 100% throughout the entire simulation period of 2013 (Figure 10ac) despite high performance in simulating discharge (Figure 10g, Table 1). Just for the hillslope contributions, the calibration by discharge only indicates possible contributions <100% during the baseflow period but shows the same uncertainty as the simulated groundwater and riparian zone contributions when the pre-monsoon and wet-up monsoon begin (Figure 10e). This strong uncertainty of the three simulated streamflow contributions despite high discharge simulation performance is a text book example of the equifinality problem (Perrin et al., 2001; Beven, 2006) that is known to result in poor prediction performance as we also found in this study when using discharge for parameter estimation only. With the experimentally derived contributions to streamflow considered in the calibration, the simulated time series of all three contributions, groundwater, hillslope and riparian zone, become more distinguishable especially during the main monsoon and the drying up period of the 2013 monsoon (Figure 10ace). We clearly see that the simulated groundwater contribution dominates discharge in the pre-monsoon and wet-up period following the observed contribution of groundwater. At the same time, the riparian zone contributions confine themselves to their observed values close to 0%. During the main monsoon and

the drying-up, the observed contributions of the hillslope are -on average- enveloped by the model simulations resulting in a substantial decrease of the groundwater contributions.

Strongly different model internal behaviour that results in almost the same discharge performance was also observed by (Seibert and McDonnell, 2002) who showed with a similar model that two completely different model setups can produce very similar discharge simulation performance. Among different types of hard and soft data, they also showed the value of observed streamflow contributions for reducing model parameter uncertainty but only focusing on two streamflow components (new water and old water) at peak discharge for six separate rainfall runoff events (McDonnell et al., 1991). In our study, we distinguish three different streamflow components temporally disaggregated over two periods that resulted in parameter uncertainty reductions that could be attributed to the respective flow and storage processes at their origin (subsection 5.2). In addition, using the monsoon year of 2014, we can show the discharge prediction performance of the model increased and simulation uncertainty decreased when the streamflow contributions are considered during parameter estimation (Figure 10h, Table 1). This is due to the improved representation of the three flow components in the model that indicate, likewise to the monsoon period in 2013, that the model could have over-estimated the contribution of the riparian zone and under-estimated the contributions of groundwater, as well as it could have miss-predicted the onset and ceasing of the hillslope contributions to discharge. Such decrease of predictive uncertainty was also revealed in other studies (Son and Sivapalan, 2007; Hartmann et al., 2017) but, to our knowledge, it has not yet been achieved using more than two experimentally separated streamflow components and not yet without accepting additional uncertainty through the incorporation of transport routines into the model.

## 6 Conclusions

The value of environmental tracers in improving the realism and prediction skills of hydrological models has been tested and proved in many previous studies. However, few studies were able to include them without adding more complexity to their models due to the conclusion of transport routines. Our study shows that, by directly comparing simulated and experimentally derived streamflow contributions, information derived from environmental tracers can be considered without adding transport routines to our model. Considering the contribution of three streamflow components, namely the hillslope, the riparian zone and groundwater, at two separate periods during a strong change of hydrological boundary conditions, we provide strong indication that it is worth considering the temporal dynamics of components that express more than just pre-event and event water in the model. Including this information in our stepwise parameter estimation procedure, we obtain increased parameter identifiability and decreased simulation uncertainty in the validation period compared to using discharge only for calibration. Incorporating the contributions of different components iteratively, we can show that they increase the identifiability of parameters related to the dynamics of their origin (e.g., the hillslope flow and storage dynamics when hillslope contributions to streamflow are considered). Considering all three observed streamflow components, we can identify all nine model parameters compared to just five parameters when using discharge only for calibration. Consequently, the uncertainty of predicted streamflow in 2014 decreases along with an increased precision of predicted streamflow components.

Our study adds to the large body of preceding work that provides evidence for the usefulness of incorporating auxiliary data into model calibration. In particular, it shows that the full potential of incorporating streamflow contributions obtained by environmental tracers has not yet been explored. On the one hand, including estimated streamflow contributions from multiple sources (not just event and pre-event water) allows enhanced improvement of the simulation of model internal processes, especially those that are seldom monitored such as hillslope contributions through subsurface stormflow (Chifflard et al., 2019). On the other hand, considering the dynamics of those streamflow contributions over time provides a more thorough distinction between realistic and unrealistic parameters combinations. We see that among the two periods that we considered, the observations for the pre-monsoon and wet-up periods are well enveloped by the simulations. But the temporal resolution of experimentally derived contributions to streamflow during the main monsoon and the drying-up period seem to be too coarse as the simulations show much higher temporal variability (while their average seems to follow the observed contributions). Hence, future efforts may involve the monitoring and integration into the model of streamflow components at a higher temporal resolution. Furthermore, separating contributions of streamflow components of different origin, our approach might be suitable for parameterization of hillslope processes in more complex and spatially distributed models at larger scales (Holmes et al., 2022; Stadnyk et al., 2013; Fan et al., 2019).

## 7 Acknowledgements

This research is a contribution to the research network *Subsurface Stormflow: A well-recognized but still challenging process in catchment hydrology research* funded by the German Research Foundation DFG (project number CH 870/5-1). The experimental work at the test catchment was supported by the German Research Foundation DFG International Research Training Group TERRECO (GRK 1565/1).

## 8 Code/Data availability

Data and model code will be published at https://github.com/KarstHub when the paper is accepted.

## 9 Author contribution

AH designed and implemented the modelling part and drafted large parts of the paper. JLPP collected the experimental data. LH applied the hydrograph separation and drafted the study site description. All authors contributed in improving and finalizing the manuscript.

## 10 Competing interests

There are no competing interests.

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
