# Peer review of "Incorporating experimentally derived streamflow contributions into model parameterization to improve discharge prediction"

_Hydrology and Earth System Sciences, 2021_

## Referee Comment (RC1)

I had the pleasure to read the paper "Incorporating experimentally derived streamflow contributions into model parameterization to improve discharge prediction" published in HESSD under https://doi.org/10.5194/hess-2021-179 by Hartmann et al. The paper is generally well-written and the quest for reducing model uncertainty and increasing model realism is of interest to the HESS readership. I found the combination of a relatively simple conceptual rainfall-runoff model constrained by observation-based water source contribution estimates worthwhile and adequate to respond the research questions asked.

Having said that, I have some comments and suggestions that I put forward for the authors to consider:

- While I appreciate the tracer sampling effort and new data from a lesser known study site, the paper structure does not reflect this at all. The mixing model and some mentioning of the tracer data used appears in the study site description even with the results of the water source contributions. I strongly suggest to separately put these into the methods and later into results assuming this data and analysis was not previously published (no reference suggests this!). Furthermore, it would be instructive to actually see some of this data to get a notion of the space-time variability, e.g. in form of a bi-variate plot since the water source estimates are crucial for the analysis. I also wonder why no throughfall end-member was included in the mixing model and if the 20% margin for $F_{HZ}$ and $F_{GW}$ used to accept/reject models is based on an uncertainty estimate of the mixing model results (none are presented in Table 1)?

- The fact that throughfall was sampled made me wonder about the importance of interception at this forested catchment and the effect it might have on the modelling since this is not included in the model structure. I also have a bit of an issue with the model structure itself and how the storage outflows are used to represent water sources: The model is essentially lumped with a vertical two-storage cascade fed by a soil reservoir that re-distributes the water for runoff generation. Now, I was thinking conceptually that all the hillslope outflow must feed into the riparian zone and from there runoff is generated that together with the groundwater flow constitutes streamflow (two end-members only). However, the latter would require a minimal semi-distributed model structure with two-storages in parallel (hillslopes draining into the riparian zone) and a third groundwater reservoir. In contrast, your HZ and GW is coming from the same source (storage V1). I would definitely appreciate some more explanations here.

- My main concern however, is that the paper falls a little short in terms of the analysis related to many assumptions that are currently not sufficiently justified. For example, the choice of the KGE statistic that clearly influences the validation of mostly low flows in 2014, which is almost unfair as there is visibly no information content in these measurements. Without tracer measurements for 2014 it almost bags the question of why 2014 was included in the first place. The threshold of KGE>0.8 to accept models seems arbitrary. All three recession constants have the same initial parameter limits, but you would certainly accept a slower response of the groundwater reservoir outflow. Or you could think of fixing the groundwater recession constant based on a Master recession curve such as suggested in work by Hrachowitz et al. On the importance of the modeler's choices. As a matter of fact there is more literature on previous work (you could potentially cite) that attempted to reduce parameter uncertainty through constraining parameters with additional information such as

tracers that did not necessarily included the need for more model complexity in terms of number of parameters. I would therefore suggest to try and test different statistics to see how they perform and apply the different criteria for model parameter selection also to the full 2million parameter sets for a more comprehensive assessment of information content. Furthermore, throughout the paper you suggest quantitative assessments of information content, uncertainty in the context of a likelihood-weighted uncertainty estimate (GLUE), parameter identifiability and sensitivity, but this was not really done. Here, I would suggest to consistently use terminology and maybe provide some quantitative analysis such as e.g. a Shannon criterion for information content and/or a sensitivity metric such as Sobol and/or a measure of the width of the likelihood-weighted uncertainty bound used for prediction. With that you more comprehensively support your interpretations and allow the reader to really assess your statements in the discussion and conclusion.

- Figure 5 is quite hard to interpret and I suggest to use a log-scale for streamflow visualization.

- There are some occasions in the paper where you wrote "be", but I think it should be "by".

For the above reasons, I would recommend major revisions before potential publication of this paper.

Sincerely,
Christian Birkel

---

## Referee Comment (RC2)

This is a well-written and interesting manuscript, although it mostly confirms what has been known for quite a while: Most hydrological models do a reasonable job in predicting water fluxes, but are pretty bad at quantifying source contributions. Rainfall-runoff modelling is "easy", because it converts a known rainfall input to an output of streamflow (with some modifications and buffering). Thus, a model may be able to predict a streamflow response, but that does not mean that the processes are inferred correctly because they cannot be constrained with the available hydrometric data (I believe that this was discussed in a commentary by James Kirchner in 2006, for example, but there are others, including those cited in the discussion section of this manuscript). This manuscript does have value, as it assesses this question rather systematically, and without increasing the dimensionality of the model substantially. However, a more thorough description of the state of the art should be included in the introduction.

I also want to echo the comment made by the first reviewer, regarding the lacking description of the tracer data. Having more information on solutes and isotopes collected is essential to be able to assess the validity of the approach. How many samples were used to quantify the end-member concentration for each period? What is the analytical uncertainty? These are important questions and would allow assigning uncertainties to the calculated end-member contributions (Table 1), which are currently lacking. I noted the reference to Payeur-Poirier, but some of the information is important enough to be repeated in this manuscript. The contributions from different end-members can be highly uncertain, but whether this uncertainty can be captured depends on the type and amount of sampling. It appears to me (but I cannot be certain since the information is not provided) that only one sample was used to quantify the end-member concentration of each period (and hence the lacking uncertainty in end-member contributions). This is rather problematic, because spatial variability in concentrations within the same end member can be large (for example, Kendall et al., 2001). Considering the changes in vegetation in the catchment (from coniferous trees in the lower parts to deciduous trees in the upper parts) it is likely that soils and weathering profiles differ, and that shallow groundwater concentrations are thus heterogeneous in the catchment.

Finally, if I understand the manuscript correctly, only the hydrometric part of the model was validated, because only streamflow data was available for the validation period, but no tracer data and thus no end-member information. If this is the case, this is a major caveat, and should be pointed out clearly in the manuscript. This means that you can tune the model to re-create the observed end-member contributions, but there is no certainty whether it can actually predict the processes occurring. (Being modellers, I am certain the authors understand the implications much better than me)

**Specific comments:**

Page 1, line 21: "using a simple framework" Can you be more specific here? This leaves the reader wondering what was done. Something like "Using a modified version of the HVB model" might be better.

Page 4, line 11: This is the only place where you mention sulphate. Why was it not used for the further analysis?

Page 4, line 17: the fourth assumption of conservative behavior is rather dubious for some of the tracers (e.g. nitrate), that may behave quite differently along different parts of the flowpaths, depending on the absence or presence of oxygen. This should be pointed out.

Page 5, line 1: (1) If you had sufficient samples to calculate a mean, you could also quantify the uncertainty of each end-member contribution. (2) I question the approach of using mean end-member contributions. Figure 5 shows highly variable end-member contributions during some of the periods, casting doubt on the validity of using mean values. Why did you not simply assign end-members to sampling times and fit to that, rather than artificially defining different periods?

Page 6, line 14: "the first storage" Is this actually correct, i.e. is it really the reservoir marked "1" in Figure 2 and not the soil storage above?

The description of the model uses the term "soil storage" quite frequently. It would be helpful to identify this (I assume this is the upper box with dashed lines) in Figure 2. Adding a short explanation in the figure caption of reservoirs 1 & 2 might be helpful.

Page 7 line 17 – page 8 line 2: Do I understand this correctly, that you cannot validate the tracer-part of the model (because there is no tracer data for the validation period), and thus are only validating the discharge model? If this is the case, please state so explicitly, as this is a major caveat.

Page 14, line 2: "and streamflow components". I assume this refers to end-member contributions. If that assumption is correct, then this directly contradicts page 8, lines 1-2 where it is stated that for 2014, no tracer data was available. Could you elaborate on what you exactly did during the validation period and how you assessed the model performance during this period?

Page 14, lines 18-19: This is not true if you account for the uncertainty in the end-member contributions.

Page 16, lines 8-10: "The uncertainty … show considerable uncertainty…" This sentence is not overly clear. Also, are you referring to uncertainty or variability here?

**References:**

Kendall, C., McDonnell, J.J. and Gu, W., 2001. A look inside 'black box'hydrograph separation models: a study at the Hydrohill catchment. Hydrological Processes, 15(10), pp.1877-1902.

Kirchner, J.W., 2006. Getting the right answers for the right reasons: Linking measurements, analyses, and models to advance the science of hydrology. Water Resources Research, 42(3).

---

## Author Comment (AC1)

Reply to comments by referee 1 (Prof. Dr. Birkel) on the manuscript "Incorporating experimentally derived streamflow contributions into model parameterization to improve discharge prediction" by Hartmann et al. The referee's comments are provided in *ITALIC*, our response in regular style.

*I had the pleasure to read the paper "Incorporating experimentally derived streamflow contributions into model parameterization to improve discharge prediction" published in HESSD under https://doi.org/10.5194/hess-2021-179 by Hartmann et al. The paper is generally well-written and the quest for reducing model uncertainty and increasing model realism is of interest to the HESS readership. I found the combination of a relatively simple conceptual rainfall-runoff model constrained by observation-based water source contribution estimates worthwhile and adequate to respond the research questions asked.*

➔ Thanks a lot for your positive assessment

*Having said that, I have some comments and suggestions that I put forward for the authors to consider:*

*- While I appreciate the tracer sampling effort and new data from a lesser known study site, the paper structure does not reflect this at all. The mixing model and some mentioning of the tracer data used appears in the study site description even with the results of the water source contributions. I strongly suggest to separately put these into the methods and later into results assuming this data and analysis was not previously published (no reference suggests this!). Furthermore, it would be instructive to actually see some of this data to get a notion of the space-time variability, e.g. in form of a bi-variate plot since the water source estimates are crucial for the analysis. I also wonder why no throughfall end-member was included in the mixing model and if the 20% margin for FHZ and FGW used to accept/reject models is based on an uncertainty estimate of the mixing model results (none are presented in Table 1)?*

➔ Thanks for this suggestion (which was also given by referee #2). Following this advice, we will re-structure the manuscript to provide a complete section of the experimental work including fieldwork, sampling, hydrochemical analysis and derivation of streamflow contributions including respective figures. After that, we will present the revised modeling part. The new structure will be set as follows (or very similar):

1. Introduction (as it is now)
2. Description of the experimental work
2.1. Test catchment: location, topography, soils, vegetation; meteorology
2.2. Hydrometric measurements (discharge)
2.3. Sampling different waters: streamflow, hillslope soil water, riparian zone soil water
2.4. Analysis of water chemistry: parameters, methods
2.5. Deriving contributions to streamflow
2.5.1. Introduction to hydrograph separation
2.5.2. Selection and characterization of end members: show data; explain that the time-invariant behavior was not given; explain that we therefore delineated four periods for which we calculated mean end member concentrations
2.5.3. Tracer time series in streamflow over sampling period

➔ A throughfall end-member was not applied because on the daily field visits during the sampling period, overland flow was never observed. We therefore did not expect throughfall, i.e. new water, to contribute directly to streamflow. As per our observations, the generation of streamflow was a result of subsurface flow processes and, thus, we only considered subsurface water sources as end members.

➔ A margin for $F_{HS}$ and $F_{GW}$ was chosen because of previous studies that highlighted the uncertainties going along with hydrograph separation (e.g., Genereux, 1998;see P8, L11-14 in the original manuscript). But since particular estimates of uncertainties of previous hydrographs separations at other sites are difficult to transfer, we found the final value of 20% through a trial-and-error procedure within realistic ranges based on the uncertainties found in previous studies. We will clarify this and discuss our value of 20% in relation to the uncertainty of the separated streamflow components, which we will state in the expanded exponential section, in the revised manuscript.

*- The fact that throughfall was sampled made me wonder about the importance of interception at this forested catchment and the effect it might have on the modelling since this is not included in the model structure. I also have a bit of an issue with the model structure itself and how the storage outflows are used to represent water sources: The model is essentially lumped with a vertical two-storage cascade fed by a soil reservoir that re-distributes the water for runoff generation. Now, I was thinking conceptually that all the hillslope outflow must feed into the riparian zone and from there runoff is generated that together with the groundwater flow constitutes streamflow (two end-members only). However, the latter would require a minimal semi-distributed model structure with two-storages in parallel (hillslopes draining into the riparian zone) and a third groundwater reservoir. In contrast, your HZ and GW is coming from the same source (storage V1). I would definitely appreciate some more explanations here.*

➔ Interception will certainly play a role in this forested catchment. However, during the monsoon period, which is our main focus and our main sampling period, the events are characterized by high intensities (daily rainfall during the monsoon period on the days with rainfall mostly between 20-125 mm). Under these circumstances, the relevance of interception in relation to the rainfall reaching the ground surface is expected to be small. Therefore, we did not represent the process of interception in the model.

➔ The model structure implemented in this study is based on the conceptual understanding of the mutual dynamics of the hillslope, the riparian zone and the groundwater. Fast flow in the hillslope is triggered when groundwater levels increase exceed heights that correspond to an effective groundwater storage of FC (maximum storage in the hillslope [mm], see Table 2 in the original

manuscript). This model process represents conceptually the impact of rising groundwater levels on lateral transmissivities that allow fast-saturated flow down the hillslope towards the riparian zone. We will clarify this in the revised version of the manuscript.

*- My main concern however, is that the paper falls a little short in terms of the analysis related to many assumptions that are currently not sufficiently justified. For example, the choice of the KGE statistic that clearly influences the validation of mostly low flows in 2014, which is almost unfair as there is visibly no information content in these measurements. Without tracer measurements for 2014 it almost bags the question of why 2014 was included in the first place. The threshold of KGE>0.8 to accept models seems arbitrary. All three recession constants have the same initial parameter limits, but you would certainly accept a slower response of the groundwater reservoir outflow. Or you could think of fixing the groundwater recession constant based on a Master recession curve such as suggested in work by Hrachowitz et al. On the importance of the modeler's choices. As a matter of fact there is more literature on previous work (you could potentially cite) that attempted to reduce parameter uncertainty through constraining parameters with additional information such as tracers that did not necessarily included the need for more model complexity in terms of number of parameters. I would therefore suggest to try and test different statistics to see how they perform and apply the different criteria for model parameter selection also to the full 2 million parameter sets for a more comprehensive assessment of information content. Furthermore, throughout the paper you suggest quantitative assessments of information content, uncertainty in the context of a likelihood-weighted uncertainty estimate (GLUE), parameter identifiability and sensitivity, but this was not really done. Here, I would suggest to consistently use terminology and maybe provide some quantitative analysis such as e.g. a Shannon criterion for information content and/or a sensitivity metric such as Sobol and/or a measure of the width of the likelihood-weighted uncertainty bound used for prediction. With that you more comprehensively support your interpretations and allow the reader to really assess your statements in the discussion and conclusion.*

➔ Thank you for these very helpful remarks. Unfortunately, only two years of observations were available with 2013 the only year, for which streamflow contributions could be calculated. Hence, we decided to use the 2014 monsoon for evaluation only. Since it is much dryer that 2013, we consider the evaluation more rigorous (at least for the simulated discharge) and still consider the rather low values of the validation $KGE_Q$ (0.02 ± 0.39) still acceptable allowing to conclude that including the streamflow contribution oin the calibration year provides more stable predictions coapred to using discharge for calibration only (which lead to a validation KGE of only -0.98 ± 1.54). We will clarify this in the revised version of the manuscript.

➔ Concerning our selection of KGE > 0.8, we relied on previous work (e.g, Hartmann et al., 2017). In order to evaluate the sensitivity of our results on this threshold, we will vary this threshold systematically, and use error measures different to KGE as also recommended by the referee. In the revised version, we will present and discuss the impact of this analysis.

➔ We will provide a more complete literature review on studies that attempted to reduce parameter uncertainty through constraining parameters with additional information such as tracers in the revised version of the manuscript.

➔ We will also double check and correct usage of terms like "information content", "sensitivity analysis", "uncertainty analysis" to provide a clear and consistent terminology throughout the paper.

*- Figure 5 is quite hard to interpret and I suggest to use a log-scale for streamflow visualization.*

➔ Ok, we will use log-scale in the revised paper.

*- There are some occasions in the paper where you wrote "be", but I think it should be "by".*

➔ We will double-check the manuscript for this type (and others).

*For the above reasons, I would recommend major revisions before potential publication of this paper.*

*Sincerely,*
*Christian Birkel*

References

Genereux, D.: Quantifying uncertainty in tracer-based hydrograph separations, Water Resour. Res., 34(4), 915–919, doi:10.1029/98WR00010, 1998.

Hartmann, A., Antonio Barberá, J. and Andreo, B.: On the value of water quality data and informative flow states in karst modelling, Hydrol. Earth Syst. Sci., 21(12), doi:10.5194/hess-21-5971-2017, 2017.

---

## Author Comment (AC2)

Reply to comments by referee 2 (anonymous) on the manuscript "Incorporating experimentally derived streamflow contributions into model parameterization to improve discharge prediction" by Hartmann et al. The referee's comments are provided in *ITALIC*, our response in regular style.

*This is a well-written and interesting manuscript, although it mostly confirms what has been known for quite a while: Most hydrological models do a reasonable job in predicting water fluxes, but are pretty bad at quantifying source contributions. Rainfall-runoff modelling is "easy", because it converts a known rainfall input to an output of streamflow (with some modifications and buffering). Thus, a model may be able to predict a streamflow response, but that does not mean that the processes are inferred correctly because they cannot be constrained with the available hydrometric data (I believe that this was discussed in a commentary by James Kirchner in 2006, for example, but there are others, including those cited in the discussion section of this manuscript). This manuscript does have value, as it assesses this question rather systematically, and without increasing the dimensionality of the model substantially. However, a more thorough description of the state of the art should be included in the introduction.*

➔ Thank you for your positive assessment. Also following the comments of referee #1, we will include a more comprehensive literature review in the introduction.

*I also want to echo the comment made by the first reviewer, regarding the lacking description of the tracer data. Having more information on solutes and isotopes collected is essential to be able to assess the validity of the approach. How many samples were used to quantify the end-member concentration for each period? What is the analytical uncertainty? These are important questions and would allow assigning uncertainties to the calculated end-member contributions (Table 1), which are currently lacking. I noted the reference to Payeur-Poirier, but some of the information is important enough to be repeated in this manuscript. The contributions from different end-members can be highly uncertain, but whether this uncertainty can be captured depends on the type and amount of sampling. It appears to me (but I cannot be certain since the information is not provided) that only one sample was used to quantify the end-member concentration of each period (and hence the lacking uncertainty in end-member contributions). This is rather problematic, because spatial variability in concentrations within the same end member can be large (for example, Kendall et al., 2001). Considering the changes in vegetation in the catchment (from coniferous trees in the lower parts to deciduous trees in the upper parts) it is likely that soils and weathering profiles differ, and that shallow groundwater concentrations are thus heterogeneous in the catchment.*

➔ Following these recommendations, and the recommendation of referee #1, a more comprehensive description of the hydrograph separation and the collected samples will be provided.

*Finally, if I understand the manuscript correctly, only the hydrometric part of the model was validated, because only streamflow data was available for the validation period, but no tracer data and thus no end-member information. If this is the case, this is a major caveat, and should be pointed out clearly in the manuscript. This means that you can tune the model to re-create the observed end-member contributions, but there is no certainty whether it can actually predict the processes occurring. (Being modellers, I am certain the authors understand the implications much better than me)*

➔ This is correct. But we consider the evaluation of the discharge prediction skills of the model still valuable because it shows that, when streamflow contributions are used for calibrations, the model obtains a better stability of discharge simulation performance and reduced uncertainty of streamflow contributions compared to the simulations by the model calibrated by discharge only. Please also refer to our response to third general comment of referee #1. We will clarify this in the revised version of the manuscript.

*Specific comments:*

*Page 1, line 21: "using a simple framework" Can you be more specific here? This leaves the reader wondering what was done. Something like "Using a modified version of the HVB model" might be better.*

➔ Specification will be added.

*Page 4, line 11: This is the only place where you mention sulphate. Why was it not used for the further analysis?*

➔ Admittedly, the original manuscript provided too condensed a description of the tracer experiments. In the revised version, we will considerably expand the section on the experimental tracer work in the test catchment, also clarifying which tracers were used for hydrograph separation.

*Page 4, line 17: the fourth assumption of conservative behavior is rather dubious for some of the tracers (e.g. nitrate), that may behave quite differently along different parts of the flowpaths, depending on the absence or presence of oxygen. This should be pointed out.*

➔ We absolutely agree and will explain the limitations of the different tracers more clearly in the revised version of the manuscript.

*Page 5, line 1: (1) If you had sufficient samples to calculate a mean, you could also quantify the uncertainty of each end-member contribution. (2) I question the approach of using mean endmember contributions. Figure 5 shows highly variable end-member contributions during some of the periods, casting doubt on the validity of using mean values. Why did you not simply assign endmembers to sampling times and fit to that, rather than artificially defining different periods?*

➔ In the revised version, we will also quantify the uncertainty of each end member and end member contribution. The repeated sampling of the end members over the sampling period revealed that the time-invariance of end members was not given. We attributed this to the pronounced change in hydrological boundary conditions during the monsoon season, with an increase of discharge by almost three orders of magnitude. The discharge response of the stream as well as hillslope soil moisture time series in three different depths clearly showed four distinct response patterns, and thus, we used these periods also for the derivation of mean end member concentrations. In the

revised version, additional figures will clearly show the different periods of the sampling period and support our usage of four periods of end member characterization.

*Page 6, line 14: "the first storage" Is this actually correct, i.e. is it really the reservoir marked "1" in Figure 2 and not the soil storage above? The description of the model uses the term "soil storage" quite frequently. It would be helpful to identify this (I assume this is the upper box with dashed lines) in Figure 2. Adding a short explanation in the figure caption of reservoirs 1 & 2 might be helpful.*

➔ Thank you for pointing this out. Indeed, soil storage and hillslope storage have been mixed a bit in the original manuscript. We will correct this and provide a more elaborated description of the model structure and its conceptual meaning will be provided (also following the comments of referee #1)

*Page 7 line 17 – page 8 line 2: Do I understand this correctly, that you cannot validate the tracer-part of the model (because there is no tracer data for the validation period), and thus are only validating the discharge model? If this is the case, please state so explicitly, as this is a major caveat.*

➔ Please see our response to the last general comment of this review and the third general comment of referee #1.

*Page 14, line 2: "and streamflow components". I assume this refers to end-member contributions. If that assumption is correct, then this directly contradicts page 8, lines 1-2 where it is stated that for2014, no tracer data was available. Could you elaborate on what you exactly did during the validation period and how you assessed the model performance during this period?*

➔ We will clarify this contradiction in the revised version of the manuscript.

*Page 14, lines 18-19: This is not true if you account for the uncertainty in the end-member contributions.*

➔ We will correct this statement.

*Page 16, lines 8-10: "The uncertainty … show considerable uncertainty…" This sentence is not overly clear. Also, are you referring to uncertainty or variability here?*

➔ We will re-phrase this sentence.

*References:*

*Kendall, C., McDonnell, J.J. and Gu, W., 2001. A look inside 'black box'hydrograph separation models: a study at the Hydrohill catchment. Hydrological Processes, 15(10), pp.1877-1902.*

*Kirchner, J.W., 2006. Getting the right answers for the right reasons: Linking measurements, analyses, and models to advance the science of hydrology. Water Resources Research, 42(3).*

---

## Author Response (AR1)

**Comment of the Associate Editor**

*Dear Authors, there are a lot of important and critical matters raised by the reviewers and so considerable developments of the paper will be necessary. It seems from your reviewer responses that these will be forthcoming and that the suggestions from the reviewers will be taken on board. However we will need a further review round and I will be looking in detail at where the manuscript is in terms of it's scientific merit and critical analyses. I look forward to the revised manuscript in due course, best wishes, Jim*

Dear editors,
Dear Jim,

After an inexcusable delay of almost one year, please find enclosed the substantially revised version of our manuscript. The revised manuscript now contains a completely new section "Experimental work and hydrograph separation" that describes the test site, the discharge measurements, the water sampling & chemical analysis, and the derivation of the end-member contributions adding 10 pages and 5 additional figures to original version of the manuscript.

In addition, following the comments of the two referees, 14 new references were added, the 2,000,000 model realisations were repeated in order to calculate six additional performance metrics for discharge (see revised methodology and newly written SI), and changes to the threshold of KGEQ and FHS/FGW were applied in order to explore the decency of our results on their rather subjective selection. Furthermore, we most of the specific comments and explained why some of them were not considered.

Please find below a detailed point-by-point response to all remarks.

Overall, we want to thank you and the two referees for their constructive and helpful remarks that helped to improve the completeness and soundness of our study. With the applied changes, and after another round of revisions, we hope that the manuscript can be considered for publication in HESS.

With best regards,

Andreas (on behalf of Luisa and Jean-Lionel

**Comments of Referee #1**

Reply to comments by referee 1 (Prof. Dr. Birkel) on the manuscript "Incorporating experimentally derived streamflow contributions into model parameterization to improve discharge prediction" by Hartmann et al. The referee's comments are provided in *ITALIC*, our response in regular style.

*I had the pleasure to read the paper "Incorporating experimentally derived streamflow contributions into model parameterization to improve discharge prediction" published in HESSD under https://doi.org/10.5194/hess-2021-179 by Hartmann et al. The paper is generally well-written and the quest for reducing model uncertainty and increasing model realism is of interest to the HESS readership. I found the combination of a relatively simple conceptual rainfall-runoff model constrained by observation-based water source contribution estimates worthwhile and adequate to respond the research questions asked.*

1. Thanks a lot for your positive assessment

*Having said that, I have some comments and suggestions that I put forward for the authors to consider:*

*- While I appreciate the tracer sampling effort and new data from a lesser known study site, the paper structure does not reflect this at all. The mixing model and some mentioning of the tracer data used appears in the study site description even with the results of the water source contributions. I strongly suggest to separately put these into the methods and later into results assuming this data and analysis was not previously published (no reference suggests this!). Furthermore, it would be instructive to actually see some of this data to get a notion of the space-time variability, e.g. in form of a bi-variate plot since the water source estimates are crucial for the analysis. I also wonder why no throughfall end-member was included in the mixing model and if the 20% margin for FHZ and FGW used to accept/reject models is based on an uncertainty estimate of the mixing model results (none are presented in Table 1)?*

2. Thanks for this suggestion (which was also given by referee #2). Following this advice, we restructured the manuscript to provide a complete section of the experimental work including fieldwork, sampling, hydrochemical analysis and derivation of streamflow contributions including respective figures. After that, we present the revised modeling part. The new structure is set as follows (or very similar):

    1.      Introduction (as it is now)
    2.      Experimental work and hydrograph separation
    2.1.   Test catchment
    2.2.   Discharge measurements
    2.3.   Water sampling and chemical analyses
    2.4.   Deriving end-member contributions to streamflow
    2.5.1. The hydrograph separation procedure
    2.5.2. Tracer time series in streamflow
    2.5.3. Characterizing the tracer signature of the end-members
    2.5.4. A switch in end-member contributions during the main monsoon period
    3.      Modeling work - Methods
    3.1.   The model

3.      The hydrological response of the catchment to the onset of the monsoon rainfalls suggested the delineation of four periods. Discharge behaved in a very specific way before, during and after the monsoon rainfalls. These four periods facilitate the description of the hydrologic response. The water chemistry data in streamflow and sampled end-members, however, did not allow a separation into these four periods. Due to the observed stream water chemistry being relatively stable during the pre-monsoon and wet-up periods (indicating one water source) and the strongly varying tracer concentrations in the end-members during these first two periods, the hydrograph separation procedure could only be applied for the main monsoon period and the drying-up. Otherwise, the assumptions for hydrograph separation would have been violated. For the pre-monsoon and wet-up period, we conclude that streamflow was dominated by groundwater contributions. The methods, results, and discussion were adapted accordingly. The methods, results, and discussion were adapted accordingly.

4.      A throughfall end-member was not applied because on the daily field visits during the sampling period, overland flow was never observed. We therefore did not expect throughfall, i.e. new water, to contribute directly to streamflow. As per our observations, the generation of streamflow was a result of subsurface flow processes and, thus, we only considered subsurface water sources as end members.

5.      A margin for $F_{HS}$ and $F_{GW}$ was chosen because of previous studies that highlighted the uncertainties going along with hydrograph separation (e.g., Genereux, 1998). Since particular estimates of uncertainties of previous hydrographs separations at other sites are difficult to transfer, we found the final value of 10% through a trial-and-error procedure within realistic ranges based on the uncertainties found in previous studies. Please note that this value was set to a stricter value in the revision compared to 20% in the original manuscript due to the consideration of only two monsoon seasons. We clarified this in the revised manuscript (see subsection 3.2 in the revised manuscript):

"The relatively large value of 10% was chosen because of the uncertainty of the end-members (as described in subsection 2.4) and previous hydrographs separations (Genereux, 1998). It's final value of 10% found by a trial-and-error-procedure, which accounts also for the uncertainties arising simplifications in our simulation model."

*- The fact that throughfall was sampled made me wonder about the importance of interception at this forested catchment and the effect it might have on the modelling since this is not included in the model structure. I also have a bit of an issue with the model structure itself and how the storage outflows are used to represent water sources: The model is essentially lumped with a vertical two-storage cascade fed by a soil reservoir that re-distributes the water for runoff generation. Now, I was thinking conceptually that all*

*the hillslope outflow must feed into the riparian zone and from there runoff is generated that together with the groundwater flow constitutes streamflow (two end-members only). However, the latter would require a minimal semi-distributed model structure with two-storages in parallel (hillslopes draining into the riparian zone) and a third groundwater reservoir. In contrast, your HZ and GW is coming from the same source (storage V1). I would definitely appreciate some more explanations here.*

6.      Interception will certainly play a role in this forested catchment. However, during the monsoon period, which is our main focus and our main sampling period, the events are characterized by high intensities (daily rainfall during the monsoon period on the days with rainfall mostly between 20-125 mm). Under these circumstances, the relevance of interception in relation to the rainfall reaching the ground surface is expected to be small. Therefore, we did not represent the process of interception in the model.

7.      The model structure implemented in this study is based on the conceptual understanding of the mutual dynamics of the hillslope, the riparian zone and the groundwater. Fast flow in the hillslope is triggered when groundwater levels exceed heights that correspond to an effective groundwater storage of $U_{GW}$ (maximum groundwater storage in the hillslope [mm], see Table 1 in the revised manuscript). This model process represents conceptually the impact of rising groundwater levels on lateral transmissivities that allow fast-saturated flow down the hillslope towards the riparian zone. This is now clarified this in subsection 3.2 of the revised version of the manuscript.

*- My main concern however, is that the paper falls a little short in terms of the analysis related to many assumptions that are currently not sufficiently justified. For example, the choice of the KGE statistic that clearly influences the validation of mostly low flows in 2014, which is almost unfair as there is visibly no information content in these measurements. Without tracer measurements for 2014 it almost bags the question of why 2014 was included in the first place. The threshold of KGE>0.8 to accept models seems arbitrary. All three recession constants have the same initial parameter limits, but you would certainly accept a slower response of the groundwater reservoir outflow. Or you could think of fixing the groundwater recession constant based on a Master recession curve such as suggested in work by Hrachowitz et al. On the importance of the modeler's choices. As a matter of fact there is more literature on previous work (you could potentially cite) that attempted to reduce parameter uncertainty through constraining parameters with additional information such as tracers that did not necessarily included the need for more model complexity in terms of number of parameters. I would therefore suggest to try and test different statistics to see how they perform and apply the different criteria for model parameter selection also to the full 2 million parameter sets for a more comprehensive assessment of information content. Furthermore, throughout the paper you suggest quantitative assessments of information content, uncertainty in the context of a likelihood-weighted uncertainty estimate (GLUE), parameter identifiability and sensitivity, but this was not really done. Here, I would suggest to consistently use terminology and maybe provide some quantitative analysis such as e.g. a Shannon criterion for information content and/or a sensitivity metric such as Sobol and/or a measure of the width of the likelihood-weighted uncertainty bound used for prediction. With that you more comprehensively support your interpretations and allow the reader to really assess your statements in the discussion and conclusion.*

8.      Thank you for these very helpful remarks. Unfortunately, only two years of observations were available with 2013 the only year, for which streamflow contributions could be calculated. Hence,

we decided to use the 2014 monsoon for evaluation only. Since it is much dryer that 2013, we consider the evaluation more rigorous (at least for the simulated discharge) and still consider the rather low values of the validation $KGE_Q$ (0.02 ± 0.34) still acceptable allowing to conclude that including the streamflow contribution during the calibration year provides more stable predictions compared to using discharge for calibration only (which lead to a validation KGE of only -0.98 ± 1.54). This is supported by the RMSE and logNSE values that indicate a lower simulation error and a better low flow performance for 2014, respectively. This is now clarified in the revised version of the manuscript at the (end of subsection 5.1).

9.      Concerning our selection of KGE > 0.8, we relied on previous work (e.g, (Hartmann et al., 2017). To analyse the sensitivity of our results to the selection of the two thresholds ($KGE_Q < 0.8$, and $F_{HS}$ and $F_{GW}$ ± 10%), we relax their values and repeat the analysis two times. Once with $KGE_Q < 0.5$ (and $F_{HS}$ and $F_{GW}$ ± 10%), and $F_{HS}$ and $F_{GW}$ ± 20% (and $KGE_Q < 0.8$). A stricter threshold was not possible as no more parameter combinations would remain. In addition, we used error measures different to KGE as also recommended by the referee. They include the Nash-Sutcliffe efficiencies, the logarithmic Nash-Sutcliffe efficiency, the Root Mean Squared Error, and the individual components of the Kling Gupta efficiency (bias, variability and correlation). These new methodological aspects are explained in the updated methods section (subsection 3.2 and the newly created supplemental information). The new results are presented in table 1 and the SI of the revised manuscript.

10.     We now provide a more complete literature review on studies that attempted to reduce parameter uncertainty through constraining parameters with additional information such as tracers in the introduction (section 1), the Experimental work section (section 2) and the Discussion (section 3) of the revised version of the manuscript. In total, 14 references were added (Klaus and Jackson, 2018; Ledesma et al., 2018; Bishop et al., 2004; Cirmo and Mcdonnell, 1997; Bergström et al., 2002; Birkel et al., 2014, 2011; Capell et al., 2012; McMillan et al., 2012; Holmes et al., 2022; Stadnyk et al., 2013; Uhlenbrook and Leibundgut, 1999; Mayer-Anhalt et al., 2022; Yang et al., 2021).

11.     We also double checked and correct usage of terms like "information content", "sensitivity analysis", "uncertainty analysis" to provide a clear and consistent terminology throughout the paper. Most explicitly, we removed the terms "information conetcnt" and "sensitivity analysis" from the text. In the new manuscript, we consistently refer to "parameter identifiability, "simulation uncertainty", and "usefulness of different data types".

*- Figure 5 is quite hard to interpret and I suggest to use a log-scale for streamflow visualization.*

12.     We uses log-scale in the revised paper (see new Fig. 10).

*- There are some occasions in the paper where you wrote "be", but I think it should be "by".*

13.     We double-checked the manuscript for this typo and removed it where necessary.

*For the above reasons, I would recommend major revisions before potential publication of this paper.*

*Sincerely,*
*Christian Birkel*

**Comments of Referee #2**

Reply to comments by referee 2 (anonymous) on the manuscript "Incorporating experimentally derived streamflow contributions into model parameterization to improve discharge prediction" by Hartmann et al. The referee's comments are provided in *ITALIC*, our response in regular style.

*This is a well-written and interesting manuscript, although it mostly confirms what has been known for quite a while: Most hydrological models do a reasonable job in predicting water fluxes, but are pretty bad at quantifying source contributions. Rainfall-runoff modelling is "easy", because it converts a known rainfall input to an output of streamflow (with some modifications and buffering). Thus, a model may be able to predict a streamflow response, but that does not mean that the processes are inferred correctly because they cannot be constrained with the available hydrometric data (I believe that this was discussed in a commentary by James Kirchner in 2006, for example, but there are others, including those cited in the discussion section of this manuscript). This manuscript does have value, as it assesses this question rather systematically, and without increasing the dimensionality of the model substantially. However, a more thorough description of the state of the art should be included in the introduction.*

1.  Thank you for your positive assessment. Also following the comments of referee #1, we included a more comprehensive literature review in the introduction (see our response 9 above).

*I also want to echo the comment made by the first reviewer, regarding the lacking description of the tracer data. Having more information on solutes and isotopes collected is essential to be able to assess the validity of the approach. How many samples were used to quantify the end-member concentration for each period? What is the analytical uncertainty? These are important questions and would allow assigning uncertainties to the calculated end-member contributions (Table 1), which are currently lacking. I noted the reference to Payeur-Poirier, but some of the information is important enough to be repeated in this manuscript. The contributions from different end-members can be highly uncertain, but whether this uncertainty can be captured depends on the type and amount of sampling. It appears to me (but I cannot be certain since the information is not provided) that only one sample was used to quantify the end-member concentration of each period (and hence the lacking uncertainty in end-member contributions). This is rather problematic, because spatial variability in concentrations within the same end member can be large (for example, Kendall et al., 2001). Considering the changes in vegetation in the catchment (from coniferous trees in the lower parts to deciduous trees in the upper parts) it is likely that soils and weathering profiles differ, and that shallow groundwater concentrations are thus heterogeneous in the catchment.*

2.  Following these recommendations, and the recommendations of referee #1, a more comprehensive description of the hydrograph separation and the collected samples is provided in the revised manuscript (see our response 2 above).

*Finally, if I understand the manuscript correctly, only the hydrometric part of the model was validated, because only streamflow data was available for the validation period, but no tracer data and thus no end-member information. If this is the case, this is a major caveat, and should be pointed out clearly in the manuscript. This means that you can tune the model to re-create the observed end-member contributions, but there is no certainty whether it can actually predict the processes occurring. (Being modellers, I am certain the authors understand the implications much better than me)*

3.      This is correct. But we consider the evaluation of the discharge prediction skills of the model still valuable because it shows that, when streamflow contributions are used for calibrations, the model obtains a better stability of discharge simulation performance and reduced uncertainty of streamflow contributions compared to the simulations by the model calibrated by discharge only. Please also refer to our response 7 to the third general comment of referee #1. This is now clarified in subsection 5.1 of the revised manuscript.

*Specific comments:*

*Page 1, line 21: "using a simple framework" Can you be more specific here? This leaves the reader wondering what was done. Something like "Using a modified version of the HVB model" might be better.*

4.      The specification was added to the abstract.

*Page 4, line 11: This is the only place where you mention sulphate. Why was it not used for the further analysis?*

5.      Admittedly, the original manuscript provided too condensed a description of the tracer experiments. In the revised version, the section on the experimental tracer work in the test catchment was considerably expanded, also clarifying (and justifying) that electric conductivity and magnesium were used as tracers for hydrograph separation (see section 2 in the revised manuscript and our response 2 to referee #1).

*Page 4, line 17: the fourth assumption of conservative behavior is rather dubious for some of the tracers (e.g. nitrate), that may behave quite differently along different parts of the flowpaths, depending on the absence or presence of oxygen. This should be pointed out.*

6.      We absolutely agree. Please refer to our previous response on expanded description of the fieldwork and selection of tracers for hydrograph separation.

*Page 5, line 1: (1) If you had sufficient samples to calculate a mean, you could also quantify the uncertainty of each end-member contribution. (2) I question the approach of using mean endmember contributions. Figure 5 shows highly variable end-member contributions during some of the periods, casting doubt on the validity of using mean values. Why did you not simply assign endmembers to sampling times and fit to that, rather than artificially defining different periods?*

7.      The repeated sampling of the end members over the sampling period revealed that the time-invariance of end members was not given (see new section 2.4 of the revised manuscript). We

attributed this to the pronounced change in hydrological boundary conditions during the monsoon season, with an increase of discharge by almost three orders of magnitude. The discharge response of the stream as well as hillslope soil moisture time series in three different depths clearly showed four distinct response patterns, and thus, we used these periods also for the derivation of mean end member concentrations. New Figure 5 (Pxx in the revised manuscript) show the uncertainty of the different end-members. But as elaborated in our response 4 to referee #1, we found the final threshold of 10% for accepting or discarding parameters set by the streamflow contributions considering typical uncertainties identified by previous studies and trial and error, which accounts also for the uncertainties arising from model simplifications. This is now clarified in the revised subsection 3.1.

*Page 6, line 14: "the first storage" Is this actually correct, i.e. is it really the reservoir marked "1" in Figure 2 and not the soil storage above? The description of the model uses the term "soil storage" quite frequently. It would be helpful to identify this (I assume this is the upper box with dashed lines) in Figure 2. Adding a short explanation in the figure caption of reservoirs 1 & 2 might be helpful.*

8.      Thank you for pointing this out. Indeed, soil storage and hillslope storage have been mixed a bit in the original manuscript. This is now corrected in the revised manuscript including an updated sketch of the model structure (Fig. 7 in the revised manuscript) indicating the soil storage explicitly:

[Figure]

*Page 7 line 17 – page 8 line 2: Do I understand this correctly, that you cannot validate the tracer-part of the model (because there is no tracer data for the validation period), and thus are only validating the discharge model? If this is the case, please state so explicitly, as this is a major caveat.*

9.      Please see our response to the last general comment of this review (and our response 3) and the third general comment of referee #1 (and our corresponding response 7).

*Page 14, line 2: "and streamflow components". I assume this refers to end-member contributions. If that assumption is correct, then this directly contradicts page 8, lines 1-2 where it is stated that for2014, no tracer data was available. Could you elaborate on what you exactly did during the validation period and how you assessed the model performance during this period?*

10. We meant that for both calibration types discharge simulation performance was inferior to 2013. We removed the confusing statement from the sentence.

*Page 14, lines 18-19: This is not true if you account for the uncertainty in the end-member contributions.*

11. Clarified in the revised version of the manuscript; "… no additional uncertainty due to additional model parameters was introduced…"

*Page 16, lines 8-10: "The uncertainty … show considerable uncertainty…" This sentence is not overly clear. Also, are you referring to uncertainty or variability here?*

12. The language issue of the sentence was removed. The sentence refers to the uncertainty of the simulated streamflow contributes. Not need to change the wording to "variability".

**References**

Bergström, S., Lindström, G., and Pettersson, A.: Multi-variable parameter estimation to increase confidence in hydrological modelling, Hydrol Process, 16, 413–421, https://doi.org/10.1002/hyp.332, 2002.

Birkel, C., Soulsby, C., and Tetzlaff, D.: Modelling catchment-scale water storage dynamics: Reconciling dynamic storage with tracer-inferred passive storage, Hydrol Process, 25, 3924–3936, https://doi.org/10.1002/hyp.8201, 2011.

Birkel, C., Soulsby, C., and Tetzlaff, D.: Developing a consistent process-based conceptualization of catchment functioning using measurements of internal state variables, Water Resour Res, 50, 3481–3501, https://doi.org/10.1002/2013WR014925, 2014.

Bishop, K., Seibert, J., Köhler, S., and Laudon, H.: Resolving the Double Paradox of rapidly mobilized old water highly variable responses in runoff chemistry, Hydrol Process, 18, 185–189, https://doi.org/10.1002/hyp.5209, 2004.

Capell, R., Tetzlaff, D., and Soulsby, C.: Can time domain and source area tracers reduce uncertainty in rainfall-runoff models in larger heterogeneous catchments?, Water Resour Res, 48, https://doi.org/10.1029/2011WR011543, 2012.

Cirmo, C. P. and Mcdonnell, J. J.: Linking the hydrologic and biogeochemical controls of nitrogen transport in near-stream zones of temperate-forested catchments: a review, Journal of Hydrology ELSEVIER Journal of Hydrology, 88–120 pp., 1997.

Genereux, D.: Quantifying uncertainty in tracer-based hydrograph separations, Water Resour Res, 34, 915–919, https://doi.org/10.1029/98WR00010, 1998.

Hartmann, A., Antonio Barberá, J., and Andreo, B.: On the value of water quality data and informative flow states in karst modelling, Hydrol Earth Syst Sci, 21, https://doi.org/10.5194/hess-21-5971-2017, 2017.

Holmes, T. L., Stadnyk, T. A., Asadzadeh, M., and Gibson, J. J.: Variability in flow and tracer-based performance metric sensitivities reveal regional differences in dominant hydrological processes across the Athabasca River basin, J Hydrol Reg Stud, 41, https://doi.org/10.1016/j.ejrh.2022.101088, 2022.

Klaus, J. and Jackson, C. R.: Interflow Is Not Binary: A Continuous Shallow Perched Layer Does Not Imply Continuous Connectivity, Water Resour Res, 54, 5921–5932, https://doi.org/10.1029/2018WR022920, 2018.

Ledesma, J. L. J., Kothawala, D. N., Bastviken, P., Maehder, S., Grabs, T., and Futter, M. N.: Stream Dissolved Organic Matter Composition Reflects the Riparian Zone, Not Upslope Soils in Boreal Forest Headwaters, Water Resour Res, 54, 3896–3912, https://doi.org/10.1029/2017WR021793, 2018.

Mayer-Anhalt, L., Birkel, C., Sánchez-Murillo, R., and Schulz, S.: Tracer-aided modelling reveals quick runoff generation and young streamflow ages in a tropical rainforest catchment, Hydrol Process, 36, https://doi.org/10.1002/hyp.14508, 2022.

McMillan, H., Tetzlaff, D., Clark, M., and Soulsby, C.: Do time-variable tracers aid the evaluation of hydrological model structure? A multimodel approach, Water Resour Res, 48, https://doi.org/10.1029/2011WR011688, 2012.

Stadnyk, T. A., Delavau, C., Kouwen, N., and Edwards, T. W. D.: Towards hydrological model calibration and validation: Simulation of stable water isotopes using the isoWATFLOOD model, Hydrol Process, 27, 3791–3810, https://doi.org/10.1002/hyp.9695, 2013.

Uhlenbrook, S. and Leibundgut, C.: Integration of tracer information into the development of a rainfall-runoff model ACCION View project World Water Development Report View project, 1999.

Yang, X., Tetzlaff, D., Soulsby, C., Smith, A., and Borchardt, D.: Catchment Functioning Under Prolonged Drought Stress: Tracer-Aided Ecohydrological Modeling in an Intensively Managed Agricultural Catchment, Water Resour Res, 57, https://doi.org/10.1029/2020WR029094, 2021.

---

## Referee Report (RR1)

**Comments on "Incorporating experimentally derived streamflow contributions into model parameterization to improve discharge prediction" by Hartmann et al.**

The authors thoroughly revised their manuscript and provided detailed responses to the comments by both reviewers. In particular, adding section 2.4 has been very helpful in understanding the geochemistry and context of the catchment. In my opinion, the manuscript can be accepted after some very minor edits.

The authors use the term "streamflow contributions" throughout the manuscript. It may not be clear that this refers to "contribution of water originating from different sources such as direct runoff from precipitation, subsurface stormflow or groundwater to total streamflow at variable flow conditions", so I wonder if this should be made clearer in the introduction.

More importantly, I am at a loss as to the meaning of the "observed streamflow contribution" (first mentioned in line 8 on page 5). How is this determined? Is this based on the tracer data? If so, this should be specified.

Figures: in most of the new figures, the different data are indistinguishable in a greyscale printout. This is particularly the case for the end member markers in Figure 5, and the timeseries in Figures 4 and 6. These could probably be improved easily without much effort by using different symbols.

Page 9, lines 12-14: "based on EC values,…, streamflow was primarily composed of groundwater". I think this reasoning may be correct, but requires some more explanation.

Page 14, line 7: I think this should be 10% rather than 20%

Page 14, lines 6 & 11: "deviation compared to the hydrograph separation" Meaning unclear, do you mean "deviation FROM the hydrograph separation"?

Page 14, line 8-10: The sentence structure appears to be tangled.

---

## Author Response (AR2)

18.01.2023

*"Dear Authors,*

*The reviewers are now happy that our considerably revised manuscript addresses many of the concern they have and both suggest we should move (with some minor corrections) to the publication stage. Thank you for your time on developing the manuscript and answering comprehensively the reviewers main points. I have just noted that I will review the final revised manuscript just to help check for any small changes required (if any). Congratulations for your paper... best wishes, Jim"*

Dear editor,

We are very pleased that the reviewers evaluated the revised version of our manuscript so positively. We implemented the changes as requested by referee #2. Below you find our point-by-point response (also see the track-changes version of the manuscript enclosed to this resubmission).

Thanks again for your patience and kind regards,

Andreas Hartmann on behalf of all co-authors

The referees' comments are provided in *ITALIC*, our response in regular style.

**Comments of Referee #1**

*The authors thoroughly revised the manuscript with more explanations on the model calibration procedure and how inclusion of tracer data helped to constrain model parameters and simulations. They now present and explain the tracer data used as end members to calculate % source contributions to streamflow, which was an additional criterion to select or reject model simulations. The latter was substantiated by additional performance criteria to assess different aspects of the simulated hydrographs, such as high flow and low flow performance, which was important for context in the case of the 2014 study year. Therefore, I am pleased to recommend publication of this paper!*

We thank Christian Birkel for his very favourable review of the revised manuscript and for his constructive and thoughtful comments on the previous manuscript version.

**Comments of Referee #2**

*The authors thoroughly revised their manuscript and provided detailed responses to the comments by both reviewers. In particular, adding section 2.4 has been very helpful in understanding the geochemistry and context of the catchment. In my opinion, the manuscript can be accepted after some very minor edits.*

We thank the referee for their positive evaluation.

*The authors use the term "streamflow contributions" throughout the manuscript. It may not be clear that this refers to "contribution of water originating from different sources such as direct runoff from precipitation, subsurface stormflow or groundwater to total streamflow at variable flow conditions", so I wonder if this should be made clearer in the introduction.*

We thank the referee for this suggestion and rephrased some sentences in the introduction (p. 3, L 3-4; p. 3, L 26-33 in the revised version of the manuscript).

*More importantly, I am at a loss as to the meaning of the "observed streamflow contribution" (first mentioned in line 8 on page 5). How is this determined? Is this based on the tracer data? If so, this should be specified.*

We agree that this may have been misleading and have replaced this expression by "experimentally derived contributions to streamflow" throughout the manuscript.

*Figures: in most of the new figures, the different data are indistinguishable in a greyscale printout. This is particularly the case for the end member markers in Figure 5, and the timeseries in Figures 4 and 6. These could probably be improved easily without much effort by using different symbols.*

Symbols were adjusted and symbol size and font size increased to improve the clarity of the figures.

*Page 9, lines 12-14: "based on EC values,…, streamflow was primarily composed of groundwater". I think this reasoning may be correct, but requires some more explanation.*

We rephrased the sentence to "Based on EC values and Mg concentrations in the stream (Figure 4) and also general streamwater chemistry, we concluded that from DOY 160 to DOY 187, i.e. also during the wet-up period, streamflow was primarily composed of groundwater" to clarify that this statement was based on our observations of streamwater chemistry (EC and Mg as presented in the manuscript and additionally also other tracer concentrations that were analyzed but not presented in this manuscript).

*Page 14, line 7: I think this should be 10% rather than 20%*

Thank you for spotting this, it was meant to be "10%". We corrected this in the revised version.

*Page 14, lines 6 & 11: "deviation compared to the hydrograph separation" Meaning unclear, do you mean "deviation FROM the hydrograph separation"?*

The sentence was rephrased accordingly.

*Page 14, line 8-10: The sentence structure appears to be tangled.*

We rephrased the sentence to "Since reliable hydrograph separation results are only available for the 2013 monsoon season, we use this year for model calibration, whereas the monsoon season of 2014, for which only discharge observations are available, was used for the validation of the model."